# TRPM7 deficiency exacerbates cardiovascular and renal damage induced by aldosterone-salt

Francisco J. Rios [1✉], Zhi-Guo Zou[1], Adam P. Harvey [1], Katie Y. Harvey[1], Livia L. Camargo [1], Karla B. Neves [1], Sarah E. F. Nichol [1], Rheure Alves-Lopes[1], Alexius Cheah[1], Maram Zahraa[1], Alexey G. Ryazanov[2], Lillia Ryazanova[3], Thomas Gudermann [4], Vladimir Chubanov [4], Augusto C. Montezano[1] & Rhian M. Touyz [1,5✉]

Hyperaldosteronism causes cardiovascular disease as well as hypomagnesemia. Mechanisms are ill-defined but dysregulation of TRPM7, a $Mg^{2+}$-permeable channel/α-kinase, may be important. We examined the role of TRPM7 in aldosterone-dependent cardiovascular and renal injury by studying aldosterone-salt treated TRPM7-deficient ($TRPM7^{+/\Delta kinase}$) mice. Plasma/tissue $[Mg^{2+}]$ and TRPM7 phosphorylation were reduced in vehicle-treated $TRPM7^{+/\Delta kinase}$ mice, effects recapitulated in aldosterone-salt-treated wild-type mice. Aldosterone-salt treatment exaggerated vascular dysfunction and amplified cardiovascular and renal fibrosis, with associated increased blood pressure in $TRPM7^{+/\Delta kinase}$ mice. Tissue expression of $Mg^{2+}$-regulated phosphatases (PPM1A, PTEN) was downregulated and phosphorylation of Smad3, ERK1/2, and Stat1 was upregulated in aldosterone-salt TRPM7-deficient mice. Aldosterone-induced phosphorylation of pro-fibrotic signaling was increased in $TRPM7^{+/\Delta kinase}$ fibroblasts, effects ameliorated by $Mg^{2+}$ supplementation. TRPM7 deficiency amplifies aldosterone-salt-induced cardiovascular remodeling and damage. We identify TRPM7 downregulation and associated hypomagnesemia as putative molecular mechanisms underlying deleterious cardiovascular and renal effects of hyperaldosteronism.

[1] Institute of Cardiovascular and Medical Sciences, BHF Glasgow Cardiovascular Research Centre, University of Glasgow, Glasgow, UK. [2] Department of Pharmacology, Rutgers Robert Wood Johnson Medical School, New Brunswick, NJ, USA. [3] Lewis Sigler Institute of Integrative Genomics, Princeton University, Princeton, NJ, USA. [4] Walther-Straub Institute of Pharmacology and Toxicology, Ludwig-Maximilians-Universität München, Munich, Germany. [5] Research Institute of McGill University Health Centre, McGill University, Montreal, QC, Canada. ✉email: Francisco.Rios@glasgow.ac.uk; rhian.touyz@mcgill.ca

Primary hyperaldosteronism is a condition of inappropriately high levels of aldosterone for sodium status that is independent of key regulators of aldosterone secretion[1,2]. It is characterized by an increased aldosterone-to-renin ratio, accounts for 10–20% of patients with hypertension, and is associated with increased cardiovascular risk[1,2]. The pathological effects of hyperaldosteronism are mediated by excessive activation of the mineralocorticoid receptor (MR) leading to volume expansion, hypokalemia, metabolic alkalosis, and cardiovascular fibrosis and injury[3]. In experimental models, high salt diet amplifies deleterious effects of aldosterone[3]. MR activation in the distal nephron results in ENaC-mediated $Na^+$ reabsorption and excretion of potassium and hydrogen ions[4,5]. In addition to perturbed $Na^+$ and $K^+$ homeostasis, hyperaldosteronism is increasingly being recognized as a cause of hypomagnesemia[6], which is independently associated with cardiometabolic disease[7,8].

Hyperaldosteronism causes intracellular $Mg^{2+}$ depletion, processes that are ameliorated by the MR antagonist, spironolactone[9,10]. Molecular mechanisms underlying aldosterone-induced cellular $Mg^{2+}$ effects are elusive, but $Mg^{2+}$ transporters may be important. Numerous $Mg^{2+}$ transporters and channels have been proposed to be functionally important in mammals, including solute carrier family 41 (SLC41) A1, SLC41A3, mitochondrial RNA splicing 2 (MRS2), and the Transient Receptor Potential ion channel subfamily M, members 6 and 7 (TRPM6/7)[11–15]. Of importance, TRPM7 is particularly relevant because it is ubiquitously expressed, it has functional significance in cardiovascular (patho)physiology and is regulated by vasoactive agents[16–18], including aldosterone, as we demonstrated previously[19]. TRPM7 is a cation channel fused to a C-terminal α-kinase domain. The channel is notably permeable to $Mg^{2+}$ but also to $Zn^{2+}$ and $Ca^{2+}$, and the TRPM7 α-kinase phosphorylates downstream targets on serine-threonine residues of proteins involved in cell proliferation, differentiation, cytoskeleton organization, contraction, and inflammatory responses[20,21]. Functional studies suggest interplay between the channel and kinase domains since $Mg^{2+}$ influx through the channel pore regulates TRPM7-kinase activity which in turn influences channel activity[22]. However, point mutations in the kinase domain were found to increase, decrease or have no effect on cation influx[22,23]. Similar discrepancies were observed in vivo because TRPM7$^{+/\Delta kinase}$ mice, which lack aa1538–1863 in the kinase domain, are hypomagnesemic[24], while mice carrying a global kinase dead mutation (K1646R) have normal plasma $Mg^{2+}$ levels[13,25]. TRPM7 has an essential and non-redundant role in cellular physiology since TRPM7 deficiency causes cell death while TRPM7 knockout mice are embryonic lethal[22,24].

We previously demonstrated the critical interplay between aldosterone and TRPM7. In aldosterone-infused mice, TRPM7 and its downstream targets are downregulated, processes ameliorated by $Mg^{2+}$ supplementation[26–28]. At the cellular level, TRPM7 kinase deficiency amplified pro-inflammatory effects of aldosterone[27]. Considering the relationship between aldosterone, TRPM7, and cellular $Mg^{2+}$ homeostasis we questioned whether aldosterone-induced cardiovascular and renal injury, hypertension, and perturbed electrolyte homeostasis involve TRPM7, by studying TRPM7-deficient mice (TRPM7$^{+/\Delta kinase}$). Additionally, we explored the role of TRPM7 in molecular mechanisms in aldosterone/salt-induced fibrosis and cardiovascular remodeling.

## Results

### Morphological features and plasma and urine biochemistry.
Body weight at the end of the study period was similar in WT and TRPM7$^{+/\Delta kinase}$ groups without any effect of treatment (Table 1). Heart size was bigger in TRPM7$^{+/\Delta kinase}$ than WT mice. Kidney size was increased by all treatments in TRPM7-deficient mice.

At baseline, plasma magnesium levels were significantly reduced in TRPM7$^{+/\Delta kinase}$ mice compared with WT controls (Table 2). Aldosterone-salt did not further reduce plasma magnesium in TRPM7$^{+/\Delta kinase}$ mice but significantly reduced levels in WT mice. Aldosterone alone reduced plasma chloride levels in both groups. Plasma levels of glucose, sodium, potassium, and phosphate were not significantly different between groups.

Aldosterone-salt caused significant albuminuria in WT and TRPM7$^{+/\Delta kinase}$ mice (Table 2). In salt-treated mice, urine albumin levels were lower compared to vehicle-treated mice. Urinary magnesium and phosphate levels were significantly decreased while sodium levels were significantly increased in TRPM7$^{+/\Delta kinase}$ mice compared with WT counterparts. Urinary electrolyte levels were significantly increased in aldosterone-salt-treated WT and TRPM7$^{+/\Delta kinase}$ mice compared with vehicle-treated counterparts.

Because TRPM7$^{+/\Delta kinase}$ treated with aldosterone-salt exhibited increased urinary sodium, we questioned whether TRPM7 influences renal sodium transporters. To address this we assessed the expression of αENaC and Na,K-ATPase in renal tissues. Aldosterone-salt increased expression of αENaC in both groups. However, Na$^+$-K$^+$-ATPase, which is a magnesium dependent enzyme was increased only in tissues from WT mice (Supplementary Fig. 1a, b).

### TRPM7 phosphorylation and tissue $Mg^{2+}$ levels.
As an index of TRPM7 activation, we assessed Ser$^{1511}$ phosphorylation in kidneys from WT and TRPM7$^{+/\Delta kinase}$ mice[23,29,30]. As shown in Fig. 1a, phosphorylation of TRPM7 was very low in TRPM7$^{+/\Delta kinase}$ mice, similar to tissues from TRPM7 kinase dead mice (R/R) that contains a point mutation K1646R. In WT mice aldosterone-salt treatment caused a significant reduction in TRPM7 phosphorylation. Expression of TRPM7 at the gene and protein levels was increased by salt and reduced in the aldosterone-salt group in WT mice (Fig. 1b and Supplementary Fig. 1c). Expression of TRPM6, another $Mg^{2+}$ transporter, was higher in kidneys from TRPM7$^{+/\Delta kinase}$ control versus WT mice (Supplementary Fig. 1d). Aldosterone-salt increased TRPM6 expression in both groups.

As demonstrated in Table 2, tissue magnesium levels were significantly lower in kidneys and hearts from TRPM7$^{+/\Delta kinase}$ versus WT mice. Aldosterone-salt reduced kidney and heart magnesium levels in WT mice similar to levels in TRPM7$^{+/\Delta kinase}$ mice.

### Blood pressure.
Baseline systolic blood pressure (SBP) at the beginning of the study was similar in WT ($119 \pm 3.1$ mmHg) and TRPM7$^{+/\Delta kinase}$ ($115 \pm 3.2$ mmHg) control mice (Supplementary Table 2). In WT animals, aldosterone-salt increased SBP at 4-weeks (Fig. 1e). Aldosterone and salt alone had no effect on SBP in WT mice. All treatments increased blood pressure in TRPM7$^{+/\Delta kinase}$ mice (Fig. 1c–e). Whereas aldosterone and aldosterone-salt blood pressure elevating effects occurred within 1 week of treatment, salt-induced hypertension was only evident after 3 weeks of treatment. From 3 weeks, blood pressure was significantly greater in aldosterone-salt-treated TRPM7$^{+/\Delta kinase}$ mice versus WT counterparts (Fig. 1e).

### Exaggerated vascular dysfunction in aldosterone-salt-treated TRPM7$^{+/\Delta kinase}$ mice.
Vascular function was assessed in mesenteric arteries by wire myography. TRPM7$^{+/\Delta kinase}$ mice exhibited increased sensitivity to ACh-induced relaxation (Fig. 2a and Supplementary Table 2). Maximal Ach-induced relaxation was significantly reduced in vessels from WT and TRPM7$^{+/\Delta kinase}$ mice treated with aldosterone and aldosterone-salt (Fig. 2b, c). Salt alone reduced Ach-induced relaxation only in vessels from TRPM7$^{+/\Delta kinase}$ mice.

**Table 1 Body, kidney, heart, and spleen weight from WT and TRPM7$^{+/\Delta kinase}$ animals.**

|  | WT, veh, $N = 10$ | WT, aldo, $n = 7$ | WT, salt, $N = 7$ | WT, aldo salt, $n = 11$ | M7+/Δ, veh, $N = 12$ | M7+/Δ, aldo, $N = 7$ | M7+/Δ, salt, $n = 7$ | M7+/Δ, aldo salt, $n = 12$ |
|---|---|---|---|---|---|---|---|---|
| Body | 13.5 ± 0.5 | 13.1 ± 0.3 | 12.9 ± 0.3 | 13.4 ± 0.4 | 13.2 ± 0.3 | 14.3 ± 0.5 | 13.2 ± 0.5 | 13.2 ± 0.3 |
| Hearts | 66.0 ± 3.0 | 63.3 ± 4.6 | 64.8 ± 4.7 | 74.5 ± 2.3* | 79.5 ± 3.8* | 76.0 ± 3.3# | 70.9 ± 6.1 | 80.4 ± 2.7 |
| Kidneys | 74.0 ± 2.1 | 85.4 ± 4.1* | 70.2 ± 3.0 | 71.4 ± 2.9 | 74.7 ± 2.1 | 98.3 ± 4.2†# | 83.4 ± 3.8†§ | 81.4 ± 2.8†‡ |
| Spleens | 56.2 ± 4.9 | 36.7 ± 3.6* | 32.8 ± 2.6* | 49.7 ± 7.4 | 30.3 ± 1.7* | 32.5 ± 1.2 | 29.4 ± 2.4 | 34.5 ± 2.7‡ |

Body (g) and organ weight (mg) were normalized by tibia length (cm). Data is expressed as mean ± SEM. *$p < 0.05$ vs WT veh; †$p < 0.05$ vs M7+/Δ veh; ‡$p < 0.05$ M7+/Δ aldo-salt vs WT aldo-salt; #$p < 0.05$ M7+/Δ aldo vs WT aldo; §$p < 0.05$ M7+/Δ salt vs WT salt.

**Table 2 Plasma and Urine analysis and tissue Mg$^{2+}$ from WT and TRPM7$^{+/\Delta kinase}$ mice.**

|  | WT, veh, $N = 9$ | WT, aldo, $n = 7$ | WT, salt, $N = 7$ | WT, aldo salt, $n = 11$ | TRPM7+/Δ, veh, $N = 11$ | TRPM7+/Δ, aldo, $N = 8$ | TRPM7+/Δ, salt, $n = 8$ | TRPM7+/Δ, aldo salt, $n = 11$ |
|---|---|---|---|---|---|---|---|---|
| **Plasma** | | | | | | | | |
| Magnesium | 0.98 ± 0.07 | 1.19 ± 0.15 | 1.19 ± 0.16 | 0.54 ± 0.12* | 0.64 ± 0.08* | 0.78 ± 0.07 | 0.81 ± 0.08 | 0.69 ± 0.08 |
| Phosphate | 1.42 ± 0.04 | 1.54 ± 0.23 | 1.70 ± 0.24 | 1.43 ± 0.01 | 1.38 ± 0.03 | 1.57 ± 0.17 | 1.62 ± 0.20 | 1.44 ± 0.02 |
| Potassium | 2.69 ± 0.26 | 2.00 ± 0.33 | 2.64 ± 0.13 | 2.41 ± 0.24 | 2.91 ± 0.30 | 2.22 ± 0.30 | 2.80 ± 0.13 | 2.78 ± 0.38 |
| Sodium | 142.2 ± 2.3 | 153.8 ± 26 | 133.9 ± 12.8 | 149.3 ± 0.59 | 141.5 ± 2.0 | 159.9 ± 17 | 148.4 ± 12.2 | 148.6 ± 0.58 |
| Calcium | 2.48 ± 0.3 | 2.84 ± 0.27 | 3.08 ± 0.33 | 2.61 ± 0.65 | 2.14 ± 0.26 | 2.14 ± 0.27 | 3.01 ± 0.27† | 2.88 ± 0.20 |
| Chloride | 57.65 ± 0.3 | 53.50 ± 3.7* | 58.50 ± 2.0 | 56.68 ± 0.22 | 57.97 ± 0.4 | 52.6 ± 1.6† | 58.53 ± 1.34 | 56.20 ± 0.18 |
| Glucose | 11.44 ± 0.6 | 12.97 ± 0.53 | 12.15 ± 1.08 | 10.72 ± 0.61 | 12.36 ± 0.4 | 12.1 ± 0.50 | 12.61 ± 1.11 | 9.43 ± 0.91 |
| **Urine** | | | | | | | | |
| Albumin | 65.64 ± 11 | 56.92 ± 14.2 | 13.01 ± 2.4* | 221.1 ± 59.2* | 78.36 ± 29 | 45.6 ± 6.05 | 17.53 ± 1.3† | 346.6 ± 92† |
| Magnesium | 11.59 ± 1.8 | 11.20 ± 1.21 | 9.97 ± 0.71 | 20.94 ± 3.8* | 6.81 ± 0.7* | 10.27 ± 2.2 | 6.36 ± 0.51 | 17.41 ± 4.4† |
| Phosphate | 18.36 ± 5.5 | 2.61 ± 0.76* | 3.76 ± 1.71* | 186.6 ± 57.9* | 6.97 ± 1.6* | 5.36 ± 12.3 | 2.91 ± 1.22† | 160.7 ± 46† |
| Sodium | 133.0 ± 48 | 135.6 ± 32.1 | 151.1 ± 15.4 | 170.1 ± 57.4 | 389 ± 103* | 203 ± 25† | 109.7 ± 30.4† | 716 ± 243†‡ |
| Potassium | 95.92 ± 15 | 96.7 ± 16.18 | 43.48 ± 11.3* | 267.3 ± 66.4* | 86.01 ± 11 | 85.36 ± 15 | 70.57 ± 11.8 | 357.5 ± 89† |
| Calcium | 2.22 ± 0.4 | 1.65 ± 0.31 | 3.57 ± 0.90 | 19.48 ± 4.6* | 2.42 ± 0.43 | 2.13 ± 0.14 | 2.58 ± 0.63 | 13.34 ± 3.2† |
| Chloride | 133.8 ± 39 | 120.0 ± 27 | 110.0 ± 19.5 | 1083 ± 256* | 139.3 ± 33 | 156.5 ± 33 | 85.63 ± 16.9 | 1106 ± 243† |
| **Tissue Mg$^{2+}$** | | | | | | | | |
| Kidneys | 1.44 ± 0.12 | 0.94 ± 0.03* | 0.76 ± 0.08* | 1.03 ± 0.07* | 0.97 ± 0.04* | 1.11 ± 0.06 | 0.81 ± 0.10 | 1.12 ± 0.06 |
| Hearts | 1.40 ± 0.05 | 1.24 ± 0.1 | 1.15 ± 0.04* | 1.11 ± 0.07* | 1.13 ± 0.05* | 1.16 ± 0.13 | 1.19 ± 0.06 | 1.26 ± 0.04 |

Plasma parameters are expressed in mmol/L. Urinary albumin was obtained as mg/L and normalized by creatinine (mmol/L). Urinary phosphate, potassium, magnesium, calcium, sodium, and chloride were obtained as mmol/L and normalized by creatinine (mmol/L). Total Mg$^{2+}$ concentration was obtained from tissue lysates (mmol/L) and normalized by protein concentration (mg/mL). Data is expressed as mean ± SE. *$p < 0.05$ vs WT veh, †$p < 0.05$ vs TRPM7+/Δ veh, ‡$p < 0.05$ WT aldo-salt vs TRPM7+/Δ aldo-salt.

Phenylephrine-induced contraction was similar in vessels from vehicle-treated WT and TRPM7$^{+/\Delta kinase}$ mice (Fig. 2d). In aldosterone-treated mice from both groups, contractile responses were amplified (Fig. 2e, f and Supplementary Table 2). Endothelium-independent vasorelaxation was evaluated by assessing SNP-induced vasorelaxation. Vessels from TRPM7$^{+/\Delta kinase}$ mice showed increased sensitivity to SNP (Fig. 2g and Supplementary Table 2). In aldosterone-salt-treated mice, SNP-induced vasorelaxation was impaired in both groups (Fig. 2h, i). Aldosterone and salt alone had no effect on SNP-induced vasorelaxation.

**Vascular remodeling and mechanics.** Vascular structural characteristics were investigated by pressure myography (Fig. 3a–i and Supplementary Fig. 2). Compared to WT, vessels from TRPM7$^{+/\Delta kinase}$ mice had thinner inner and outer diameters (Supplementary Fig. 2) and cross-sectional area, with no changes in wall:lumen ratio (Fig. 3a, d). These findings suggest possible hypotrophic remodeling. In WT animals, aldosterone-salt treatment caused an increase in wall:lumen ratio (Fig. 3e, h, i). However, vessels from aldosterone- or salt-treated TRPM7$^{+/\Delta kinase}$ mice exhibited reduced wall:lumen ratio (Fig. 3f, h, i).

Vascular distensibility was reduced by aldosterone-salt treatment in vessels from both WT and TRPM7$^{+/\Delta kinase}$ mice as evidenced by a significant leftward shift of the stress:strain curve (Fig. 3j–m).

**Renal inflammation in aldosterone-salt-treated TRPM7$^{+/\Delta kinase}$ mice.** Increased total inflammatory infiltrate (CD45+ cells) was found in kidneys from TRPM7$^{+/\Delta kinase}$ control mice and WT mice treated with aldosterone and aldosterone-salt (Supplementary Fig. 3a). Further characterization showed increased macrophages (F4/80+ cells) in WT mice by all treatments and in TRPM7$^{+/\Delta kinase}$ controls, which was further increased by salt and aldosterone-salt (Supplementary Fig. 3b). Aldosterone-salt enhanced CD11c/CD206 expression (pro-inflammatory M1 macrophages) which was potentiated in TRPM7$^{+/\Delta kinase}$ mice (Supplementary Fig. 3c). Regarding lymphocytes, comparable frequencies of total T cells were found in kidneys from both groups, but TRPM7$^{+/\Delta kinase}$ mice exhibited increased CD4+ T lymphocytes, whereas in WT animals, these parameters were observed only in treated groups. Furthermore, increased cytotoxic CD8+ T lymphocytes was found only in kidneys from treated TRPM7$^{+/\Delta kinase}$ mice (Supplementary Fig. 3d–f). Changes in the immune cell profile was also observed in spleens (Supplementary Fig. 3g–k), which exhibited reduced mass in TRPM7$^{+/\Delta kinase}$ vehicle controls and in WT animals treated with aldosterone or salt (Table 1).

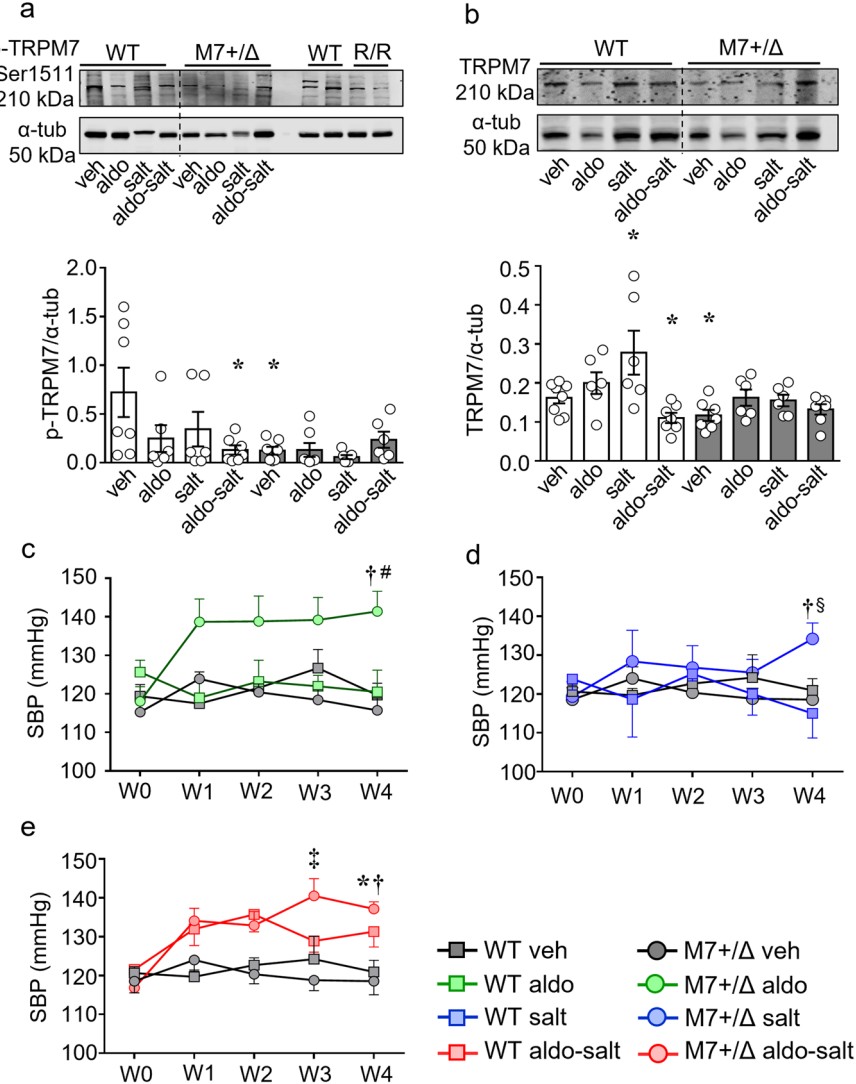

**Fig. 1 Aldosterone-salt treatment reduces TRPM7 expression and phosphorylation and increased blood pressure in WT and TRPM7$^{+/\Delta kinase}$ animals.**
Total tissue lysate from kidneys was analyzed for **a** phospho-TRPM7 (Ser1511) and **b** TRPM7 expression by immunoblotting and normalized to α-tubulin ($n = 6$–7/group). WT, white bars and TRPM7$^{+/\Delta kinase}$ (M7+/Δ), gray bars. Tissues from TRPM7 kinase-dead mice (TRPM7R/R) were used as positive control of the anti-phospho-TRPM7 (Ser1511). Graph data are presented as mean ± SEM. One-way ANOVA followed by Dunnett's multiple comparisons test were used for statistical analysis. *$P < 0.05$ vs WT veh. Blood pressure values in WT (veh, gray squares) and TRPM7$^{+/\Delta kinase}$ (M7+/Δ) (veh, gray circles) treated with **c** aldosterone (aldo, green), **d** high salt (salt, blue) or **e** aldosterone-salt (aldo-salt, red). Data are presented as mean ± SEM. $N$ numbers: WT (veh=14; aldo=10; salt=7; aldo-salt=13), M7+/Δ (veh=17; aldo=9; salt=9; aldo-salt=15). Two-way ANOVA followed by Bonferroni post hoc test were used for statistical analysis. *$p < 0.05$ vs WT veh; †$p < 0.05$ vs M7+/Δ veh; ‡$p < 0.05$ WT aldo-salt vs M7+/Δ aldo-salt.

**Cardiovascular and renal fibrosis induced by aldosterone-salt is amplified in TRPM7$^{+/\Delta kinase}$ mice.** Collagen deposition in hearts and kidneys was significantly increased in aldosterone-salt-treated WT mice versus vehicle-treated counterparts (Fig. 4 and Supplementary Figs. 4–15). Similar effects were observed in TRPM7$^{+/\Delta kinase}$ mice, but responses were amplified compared to WT mice. Collagen deposition was also found in aortas from aldosterone-salt-treated TRPM7$^{+/\Delta kinase}$ mice. In basal conditions, cardiac collagen content was already increased in TRPM7$^{+/\Delta kinase}$ versus WT mice.

Having demonstrated exaggerated aldosterone-salt-induced fibrosis in TRPM7-deficient mice, we next interrogated putative Mg$^{2+}$-sensitive signaling pathways associated with these processes especially in the heart, where effects were most pronounced. Protein phosphatase magnesium-dependent 1A (PPM1A), a Mg$^{2+}$ dependent protein and a negative regulator of Smad3, which is associated pro-inflammatory/profibrotic pathways[31,32] was significantly downregulated in cardiac tissues from all treated TRPM7$^{+/\Delta kinase}$ groups (Fig. 5a). Expression of PPM1A in kidneys and aortas was also decreased by aldosterone-salt in TRPM7-deficient mice (Fig. 5b, c). Another Mg$^{2+}$-sensitive negative regulator of cell function is phosphatase and tensin homolog (PTEN)[33], which was significantly decreased in cardiac tissues from treated TRPM7-deficient mice (Fig. 5d). Aldosterone and aldosterone-salt also reduced PTEN expression in kidneys from TRPM7-deficient mice (Fig. 5e). PTEN was not influenced by any treatments in WT mice. Associated with decreased cardiac PPM1A and PTEN in aldosterone-salt-treated TRPM7$^{+/\Delta kinase}$ mice was increased phosphorylation of Smad3, ERK1/2 and Stat1 (Fig. 6a–c) and increased expression of fibronectin gene (*FN1*), effects that were enhanced compared with WT counterparts (Supplementary Fig. 16). Expression of TGFβ1 (Fig. 6d), IL-11 and IL-6 (Supplementary Fig. 17), mediators involved in cardiovascular fibrosis and inflammation,

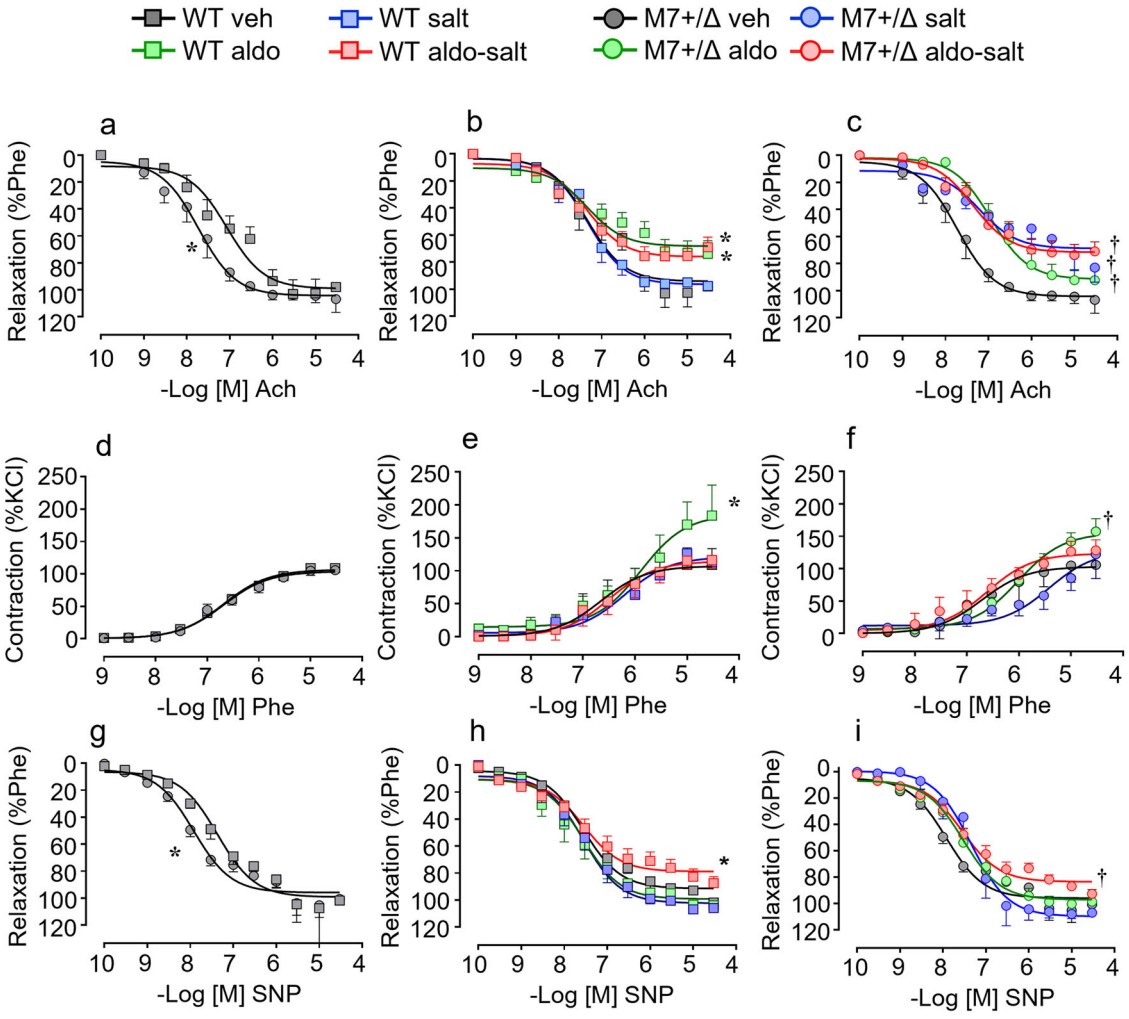

**Fig. 2 Vascular function in WT and TRPM7$^{+/\Delta kinase}$ mice.** Animals WT (squares) and TRPM7$^{+/\Delta kinase}$ (M7+/Δ, circles) were treated with aldosterone (aldo, green), high salt (salt, blue) or aldosterone-salt (aldo-salt, red). Vascular function was assessed in mesenteric arteries by wire myograph. **a–c** vascular relaxation to acetylcholine (Ach); **d–f** vascular contractility to phenylephrine (Phe); **g–i** vascular relaxation to sodium nitroprusside (SNP). $N$ numbers: WT (veh=9–14; aldo=7–10; salt=7; aldo-salt=11–13), M7+/Δ (veh=10–13; aldo=8; salt=7–8; aldo-salt=12–15). Data are presented as mean ± SEM. Concentration-response data were analyzed by determining EC50 and maximal response (Emax) values from experimental data fitted to a four-parameter logistic function against the null hypothesis (similar dataset). Null hypothesis was rejected when $p < 0.05$. *$P < 0.05$ vs WT veh; †$p < 0.05$ vs M7+/Δ veh.

were increased in hearts from TRPM7-deficient mice and aldosterone-salt treated WT mice.

**Oxidative stress in aldosterone-salt treated mice.** Supplementary Fig. 18a–c, demonstrates expression profiles of Nox isoforms in treated and untreated WT and TRPM7$^{+/\Delta kinase}$ mice. Expression of Nox1, but not Nox2 or Nox4, was increased in untreated TRPM7$^{+/\Delta kinase}$ mice similar to that in treated WT mice, suggesting upregulation of Nox1 in basal conditions in mice deficient in TRPM7. In aldosterone- and salt-treated TRPM7$^{+/\Delta kinase}$ mice, expression of Nox4 was reduced. Since Nox4 has been implicated to be cardioprotective[34], reduced expression in TRPM7-deficient mice may represent loss of protective mechanisms leading to cardiac injury in these animals. Irreversible protein oxidation of peroxiredoxin (Prx-SO3) and protein tyrosine phosphatases (PTP), was increased in hearts from WT mice treated with aldosterone or salt (Supplementary Fig. 18e, f).

**Aberrant signaling in cardiac fibroblasts from TRPM7$^{+/\Delta kinase}$ mice is magnesium-dependent.** Considering that fibroblasts are the effector cells in cardiac fibrosis we cultured fibroblasts from hearts from WT and TRPM7$^{+/\Delta kinase}$ mice and evaluated molecular and cellular effects of aldosterone and salt in the absence and presence of extracellular Mg$^{2+}$ supplementation. Corroborating the in vivo results observed in cardiac tissue, treatment-induced phosphorylation of ERK1/2, Smad3 and Stat1 was greater in cells from TRPM7$^{+/\Delta kinase}$ mice versus WT counterparts, effects that were significantly reduced by Mg$^{2+}$ supplementation (Fig. 7a–c). Compared with WT cells, TRPM7$^{+/\Delta kinase}$ cells exhibited increased expression of TGFβ1, IL-11, and IL-6, effects that were amplified by aldosterone and salt (Supplementary Fig. 19). Mg$^{2+}$ supplementation reduced the effects of aldosterone-salt stimulated TGFβ1 and IL-6 in TRPM7$^{+/\Delta kinase}$ cells.

To investigate the functional significance of Mg$^{2+}$ in TRPM7 effects, we assessed cell proliferation using the CFSE assay (Fig. 7e). The proliferation rate was significantly reduced in cardiac fibroblasts from TRPM7$^{+/\Delta kinase}$ mice compared with WT cells. This was further affected by aldosterone. Mg$^{2+}$ supplementation normalized fibroblast proliferation in TRPM7$^{+/\Delta kinase}$ mice.

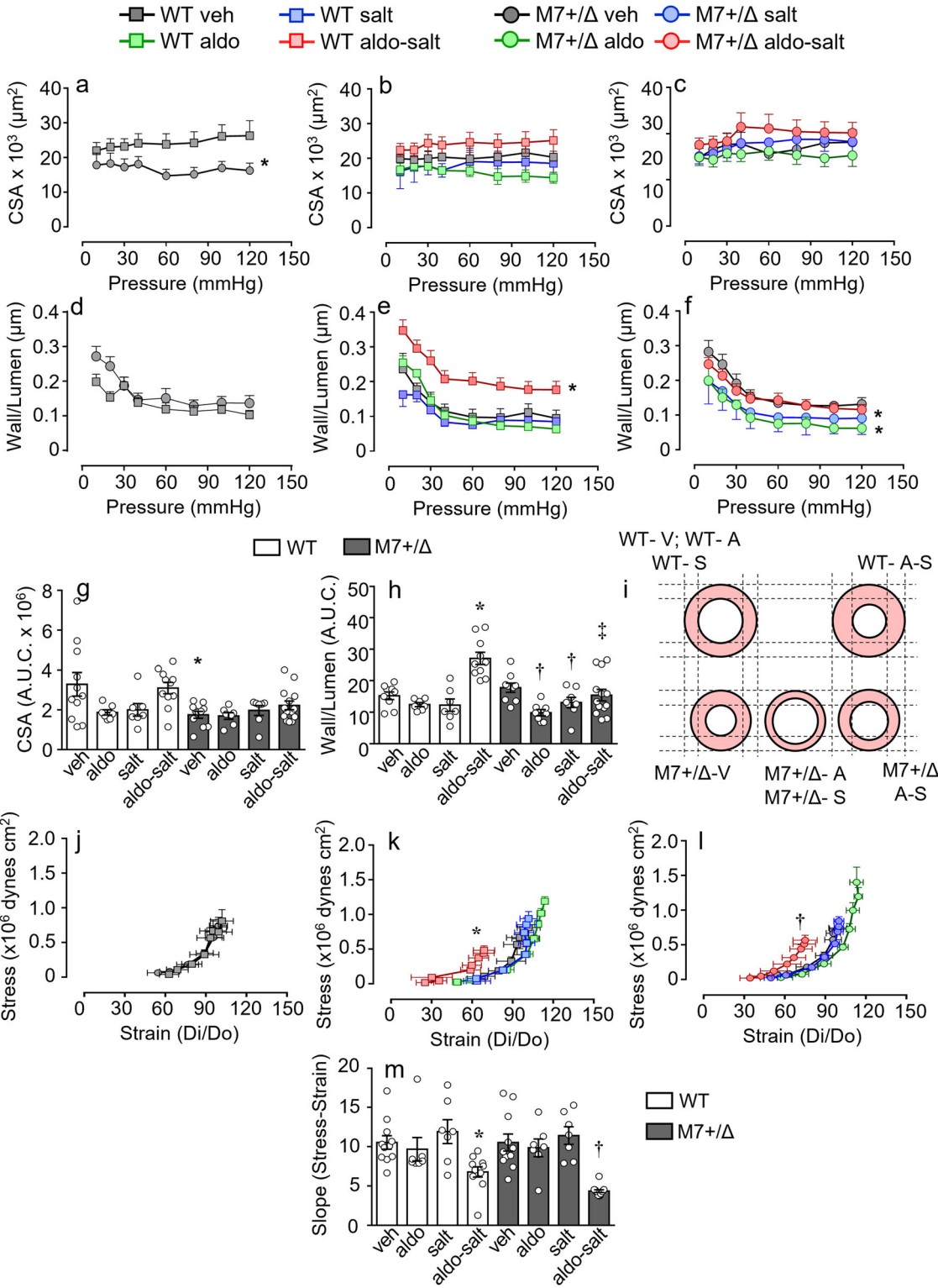

**Fig. 3 Vascular remodeling and mechanical properties in WT and TRPM7+/Δkinase mice.** Animals WT (squares) and TRPM7+/Δkinase (M7+/Δ) (circles) were treated with aldosterone (aldo, green), high salt (salt, blue) or aldosterone-salt (aldo-salt, red). Vascular structure and mechanical properties were assessed in pressurized mesenteric arteries at increasing intraluminal pressure (10–120 mmHg) in calcium-free conditions. **a**–**c**, **g** cross-sectional area (CSA); **d**–**f**, **h** wall to lumen ratio; **i** schematic figure of the changes in vascular structure. Mechanical properties are presented as **j**–**l** vascular stress–strain curves and **m** slope of the curves. N numbers: WT (veh=11; aldo=7; salt=7; aldo-salt=11), M7+/Δ (veh=11; aldo=7; salt=7; aldo-salt=14) Data are present as mean ± SEM of Area under the curve (A.U.C.) of WT (white bars) and M7+/Δ (gray bars). One-way ANOVA followed by Dunnett's multiple comparisons test were used for statistical analysis (**a**–**h**). Differences in vascular mechanics were assessed using values of slope of sress–strain curves **j**–**l** that were analized using one-way ANOVA followed by Dunnett's multiple comparisons test (**m**). *$P < 0.05$ vs WT veh; †$p < 0.05$ vs M7+/Δ veh. ‡M7+/Δ aldo-salt vs WT aldo-salt.

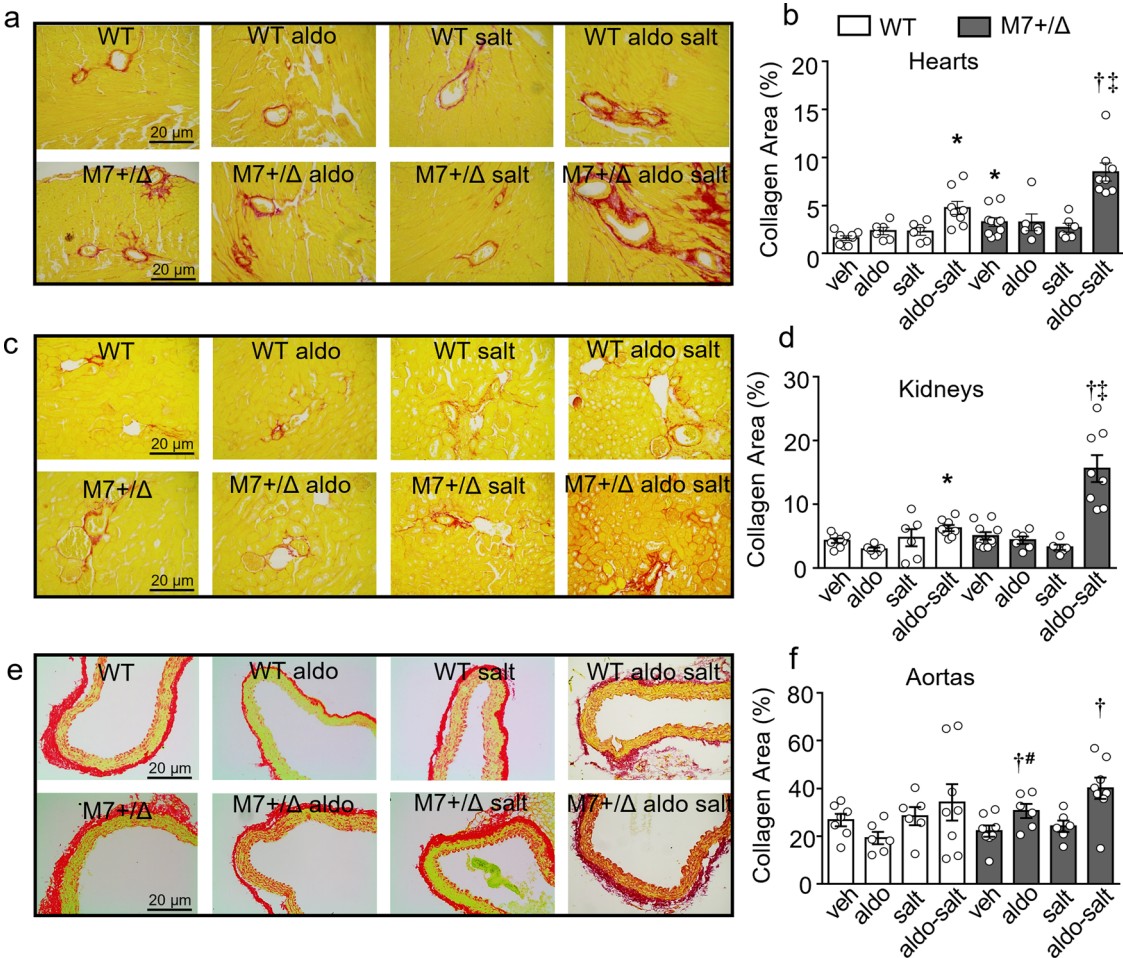

**Fig. 4 Increased collagen deposition in hearts, kidneys, and aortas from WT and TRPM7+/Δkinase mice treated with aldosterone and salt.** Tissues from WT and TRPM7+/Δkinase (M7+/Δ) mice were stained with picro-sirius red. Collagen content was assessed in bright field microscopy (scale bar 20 μm) and analyzed using the Image J software. **a**, **b** hearts, **c**, **d** kidneys, and **e**, **f** aortas. N numbers: WT (veh=7-8, aldo=6, salt=6, aldo-salt=8); M7+/Δ (veh=7-8, aldo =6, salt =6, aldo-salt =8). Data are expressed as representative images and mean ± SEM of % of affected area. WT: white bars; M7+/Δ: gray bars. One-way ANOVA followed by Dunnett's multiple comparisons test were used for statistical analysis. *P < 0.05 vs WT veh; †p < 0.05 vs M7+/Δ veh; ‡p < 0.05 WT aldo-salt vs M7+/Δ aldo-salt; # M7+/Δ aldo vs WT aldo.

## Discussion

Major findings from our study demonstrate that TRPM7 deficiency exacerbates aldosterone-salt-induced cardiovascular and renal injury, vascular dysfunction, and perturbed $Mg^{2+}$ and electrolyte homeostasis. These processes are associated with abnormal profibrotic signaling pathways. In particular, we show that $Mg^{2+}$-regulated proteins, PPM1A and PTEN, are downregulated while Smad3, ERK1/2, and Stat1 are upregulated by aldosterone-salt in hearts from TRPM7-deficient mice. These effects are mediated, in part, by $Mg^{2+}$ deficiency, because $Mg^{2+}$ supplementation ameliorated Smad3, ERK1/2, and Stat1 phosphorylation induced by aldosterone and salt in TRPM7-deficient cardiac fibroblasts (Fig. 8). Together, our study demonstrates that cardiovascular injury mediated by aldosterone-salt is associated with decreased TRPM7 activity and that TRPM7 deficiency amplifies injurious effects, partially through perturbed cellular $Mg^{2+}$ homeostasis. We identify TRPM7 downregulation as a putative mechanism underlying deleterious cardiovascular effects of hyperaldosteronism.

Aldosterone is increasingly being recognized as a magnesiuric hormone that also controls intracellular $Mg^{2+}$ homeostasis[3,10,35–37]. In addition, aldosterone is regulated by $Mg^{2+}$[38]. This is supported by the findings that: (i) infusion of $Mg^{2+}$ suppresses plasma aldosterone levels in humans, (ii) exposure to high $Mg^{2+}$ decreases aldosterone production in rat zona glomerulosa cells,

and (iii) experimental $Mg^{2+}$ deficiency stimulates aldosterone production[39,40]. To explore molecular mechanisms associated with the interplay between aldosterone, $Mg^{2+}$ and cardiovascular injury we focused on TRPM7, the ubiquitously expressed $Mg^{2+}$ transporter. We found hypomagnesemia and reduced phosphorylation of TRPM7-kinase in WT mice treated with aldosterone and salt, phenomena observed in TRPM7+/Δkinase animals in basal unstimulated conditions. Using an antibody specific to the upstream portion of the truncated kinase in TRPM7+/Δkinase, we found that aldosterone and salt also reduce the expression of the TRPM7 channel domain to the same levels observed in TRPM7-deficient animals, suggesting that aldosterone mimics, in part, the phenotype of TRPM7 deficiency. Additionally, mice treated with aldosterone and salt exhibited urinary $Mg^{2+}$ and $K^+$ wasting, similar to observations in patients with hyperaldosteronism[2,5]. Processes, whereby aldosterone causes these electrolyte abnormalities, may relate to altered renal and/or intestinal TRPM7 activity, because TRPM7, together with TRPM6, control $Mg^{2+}$ reabsorption in the kidney and gastrointestinal system[11,13,41]. The hypomagnesemic effects of aldosterone have important clinical significance and may contribute to cardiac arrythmias in hyperaldosteronism. In particular long QT syndrome, typically associated with hypomagnesemia and treated with $Mg^{2+}$, has been described in patients with hyperaldosteronism[42,43].

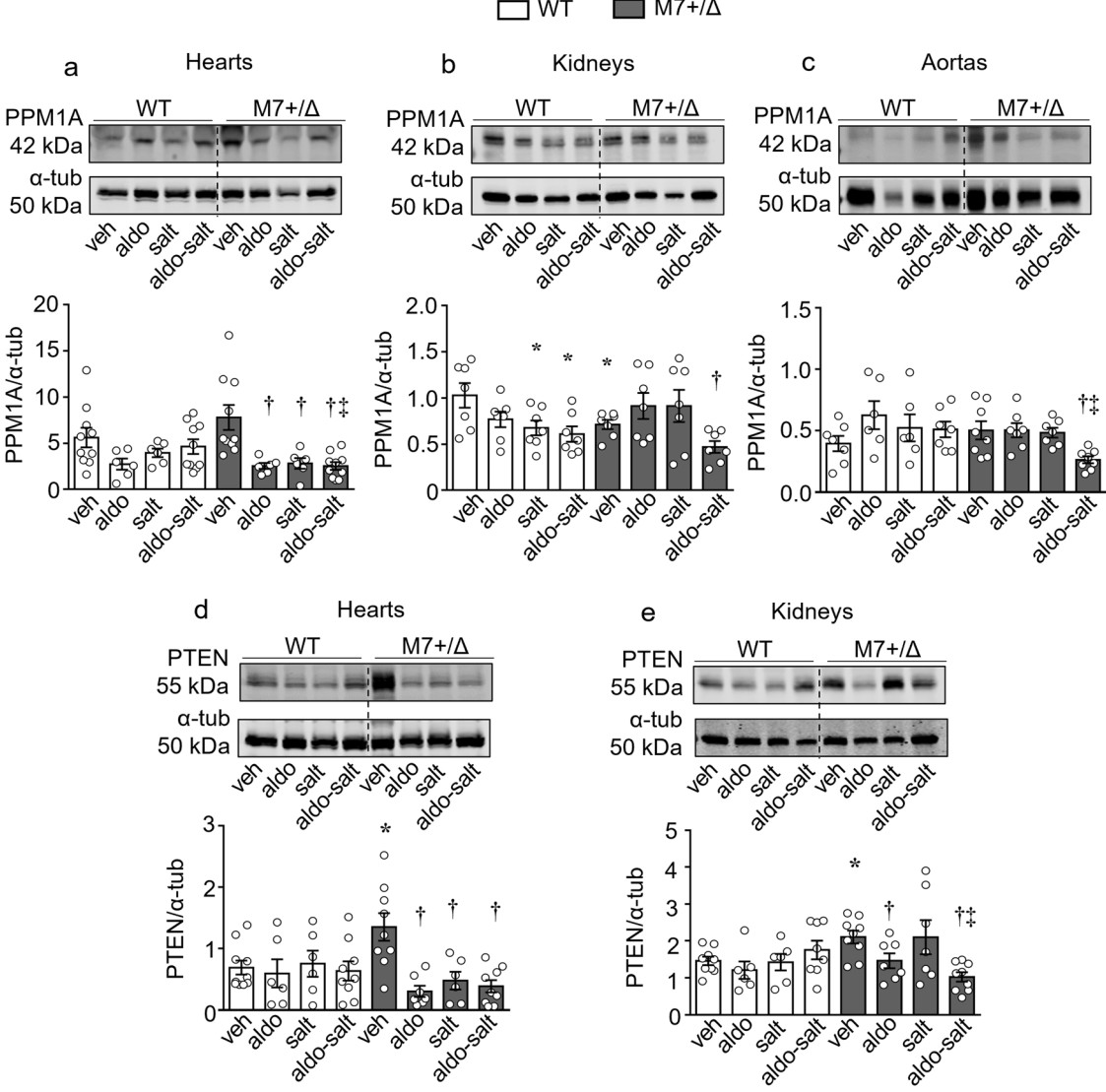

**Fig. 5 TRPM7 deficiency is associated with reduced expression of the phosphatases PPM1A and PTEN.** Tissues from mice WT (white bars) and TRPM7$^{+/\Delta kinase}$ (gray bars, M7+/Δ) were investigated for the expression of PPM1A in **a** hearts ($n = 6$–$10$/group), **b** kidneys ($n = 7$/group), and **c** aortas ($n = 6$–$8$/group). PTEN expression was investigated in **d** hearts ($n = 6$–$10$/group) and **e** kidneys ($6$–$9$/group). Protein expression was normalized to α-tubulin. Data are expressed as mean ± SEM and representative figures. One-way ANOVA followed by Dunnett's multiple comparisons test were used for statistical analysis. *$P < 0.05$ vs WT veh; †$p < 0.05$ vs M7+/Δ veh; ‡$p < 0.05$ WT aldo-salt vs M7+/Δ aldo-salt.

In addition to Mg$^{2+}$ wasting, aldosterone-salt-treated TRPM7$^{+/\Delta kinase}$ mice exhibited marked natriuresis. Exact reasons for this are unclear. Since we did not perform metabolic studies, we cannot estimate the exact salt intake for each animal, which is a limitation of our data. However, our findings suggest that TRPM7 kinase may play an important role in renal sodium handling. To explore this possibility, we examined expression profiles of major sodium transporters in the kidney specifically ENaC and Na,K-ATPase. Aldosterone-salt similarly increased expression of ENaC in WT and TRPM7-deficient mice. However, aldosterone-salt upregulated Na,K-ATPase in WT mice but not in treated TRPM7$^{+/\Delta kinase}$ mice. Na,K-ATPase, which is a Mg$^{2+}$-sensitive ATPase, controls intracellular Na$^+$ concentration by promoting Na$^{2+}$ reabsorption in the basolateral surface of the nephron[38]. At physiological concentrations of Mg$^{2+}$, Na,K-ATPase pumps three Na$^+$ out of the cell in exchange for two K$^+$ entering the cell. Reduced intracellular [Na$^+$] forces the Na$^+$ gradient through the luminal surface, thereby increasing Na$^+$ reabsorption[38]. In the context of hypomagnesemia in

TRPM7$^{+/\Delta kinase}$ mice, these effects may be attenuated resulting in increased natriuresis. These findings suggest important interactions between TRPM7 kinase and renal sodium handling and warrant further investigation into exact mechanisms underlying these phenomena.

The negative relationship between TRPM7 and blood pressure was previously highlighted in various models of hypertension, where we found decreased TRPM7 expression in cardiovascular and renal tissues from spontaneously hypertensive rats and in Ang II and aldosterone-treated mice[13,26,44]. In previous studies, we showed that aldosterone induced cardiovascular damage and reduced TRPM7 gene expression by magnesium-dependent mechanisms[26]. We also demonstrated that Ang II-infused TRPM7$^{+/\Delta kinase}$ mice have severe hypertension and reduced activation of the TRPM7-kinase target proteins annexin-A1 and calpain II[44]. Here we advance the notion and show that TRPM7$^{+/\Delta kinase}$ mice are more sensitive to blood pressure-elevating effects of aldosterone than WT mice, which only develop hypertension when salt is added. While vascular function

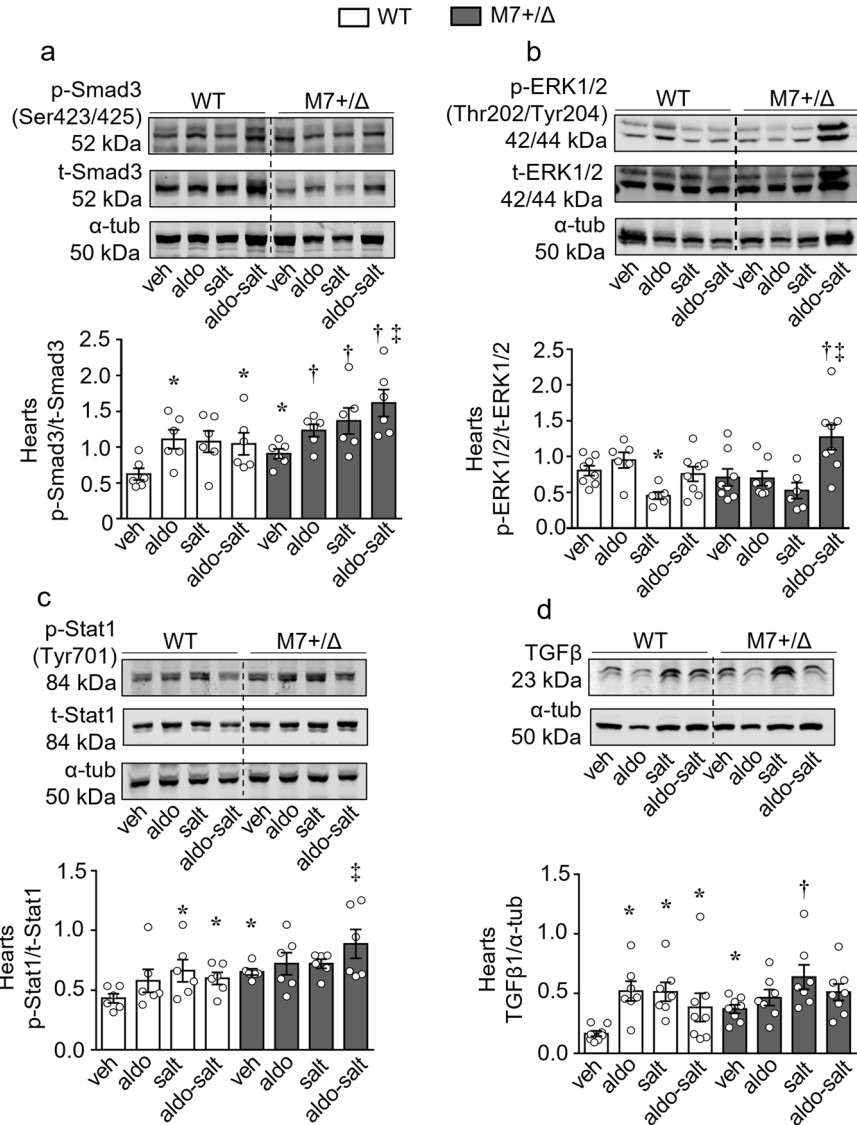

**Fig. 6 TRPM7 deficient mice exhibit increased activation of Smad3, ERK1/2 and Stat1 and expression TGFβ1.** Heart tissues from WT (white bars) and TRPM7$^{+/\Delta kinase}$ (gray bars, M7+/Δ) were investigated for the expression of **a** phospho-Smad3 (Ser423/425), **b** phospho-ERK1/2 (Thr202/Tyr204), **c** phospho-Stat1(Tyr701) and **d** TGFβ1. Protein expression was normalized by total forms of Smad3, ERK1/2, Stat1 or α-tubulin. Data are expressed as mean ± SEM and representative figures ($n = 6$–8/group). One-way ANOVA followed by Dunnett's multiple comparisons test were used for statistical analysis. *$P < 0.05$ vs WT veh; †$p < 0.05$ vs M7+/Δ veh; ‡$p < 0.05$ WT aldo-salt vs M7+/Δ aldo-salt; ‡$p < 0.05$ WT aldo-salt vs M7+/Δ aldo-salt.

(decreased vasorelaxation and increased contraction) was similarly impaired in aldosterone-salt-treated WT and TRPM7$^{+/\Delta kinase}$ mice, structural changes differed between groups. Aldosterone-salt induced significant hypertrophic remodeling in WT mice, whereas vessels from TRPM7$^{+/\Delta kinase}$ mice exhibited reduced wall/lumen ratio under aldosterone or salt treatment, effects that were further altered with aldosterone-salt treatment. These findings suggest that TRPM7-deficency is associated with impaired vascular adaptation, which may be deleterious in the setting of hypertension. Processes contributing to the thin vascular media and hypotrophic remodeling in TRPM7$^{+/\Delta kinase}$ mice are still unclear, but abnormal VSMC proliferation, apoptosis, and growth may be important[45]. Supporting this, we showed reduced proliferation of TRPM7$^{+/\Delta kinase}$ fibroblasts, which was further attenuated by aldosterone, phenomena that were normalized by Mg$^{2+}$ supplementation.

TRPM7 plays a crucial role in vasculogenesis and cell growth, since TRPM7 knockout mice die at embryonic day 7.5[24], which coincides with the formation of the first primitive vessels[46].

Additionally, pharmacological or genetic inhibition of TRPM7 and intracellular Mg$^{2+}$ depletion, inhibit VSMC proliferation and migration by mechanisms dependent on ROS production and ERK1/2 phosphorylation[47]. Previous studies showed that the C-terminal kinase domain of TRPM7 can be cleaved, translocate to the nucleus, and induce epigenetic modifications, in a cell-type-specific manner[48]. Therefore, it is possible that these processes may also occur in cardiovascular damage induced by aldosterone and salt. These factors may contribute to the altered structure of resistance arteries in TRPM7$^{+/\Delta kinase}$ mice.

Aldosterone is a potent inducer of inflammation and fibrosis, which lead to arterial stiffness and tissue damage[1,49]. Aldosterone induces inflammation by increasing the expression of ICAM-1, ROS production, IL-6 production, and activation of NFκB[27]. We showed that aged TRPM7$^{+/\Delta kinase}$ mice exhibit increased systemic inflammatory responses and cardiac fibrosis by mechanisms related to macrophage activation and Mg$^{2+}$ deficiency[28]. Investigating inflammatory cell infiltrates in kidneys from TRPM7$^{+/\Delta kinase}$ mice, we confirmed the pro-inflammatory phenotype in these animals.

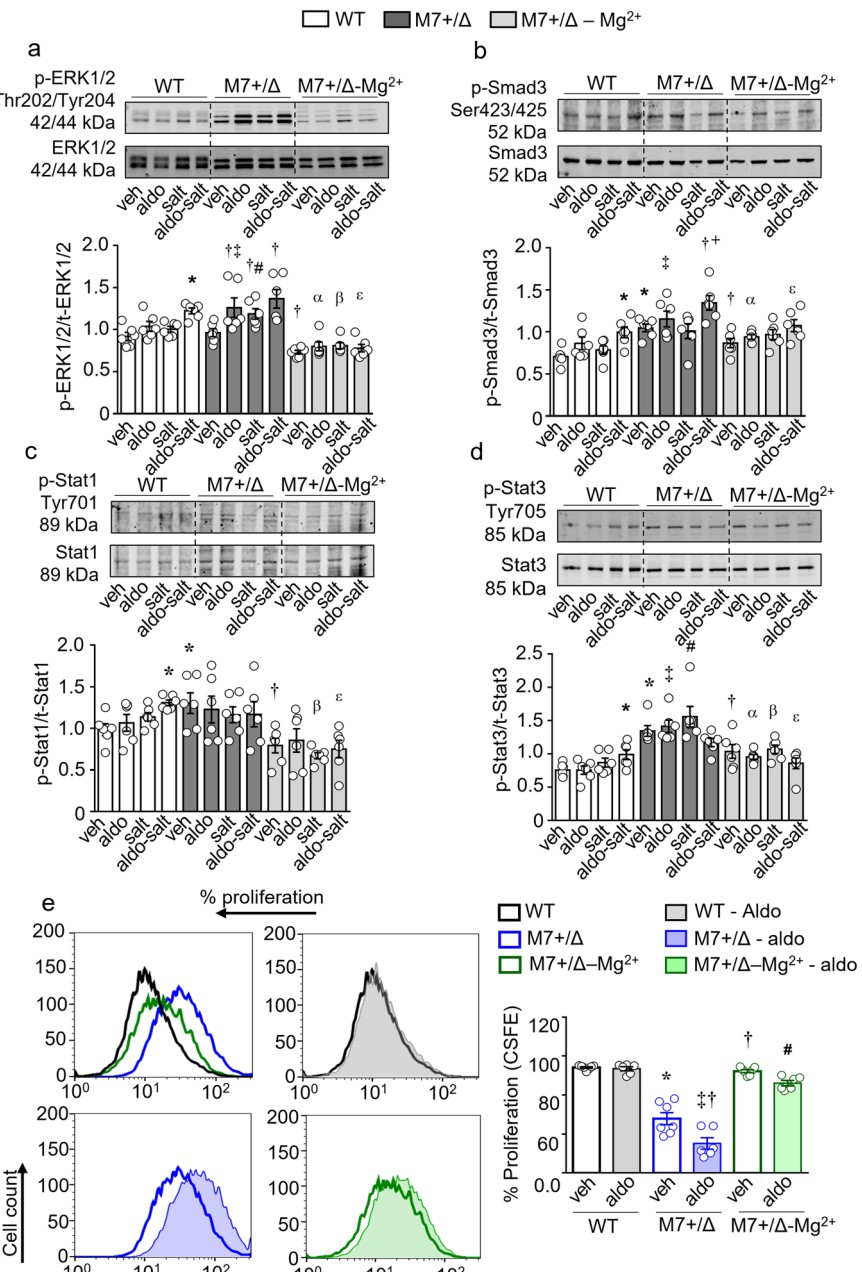

**Fig. 7 Cardiac fibroblasts from TRPM7$^{+/\Delta kinase}$ exhibit aberrant activation and proliferation by Mg$^{2+}$ dependent mechanisms.** Cardiac fibroblasts were isolated from WT and TRPM7$^{+/\Delta kinase}$ (M7+/Δ) animals. Part of the cells from M7+/Δ animals were constantly treated with 10 mM of MgCl$_2$. 24 h before the experiments, medium was changed to 1% FBS and part of the cells were treated with NaCl 40 mM or osmotic control Choline Chloride (40 mM). Cells were stimulated with aldosterone (10$^{-7}$ mmol/L) for 10 min. Expression of **a** phospho-ERK1/2 (Thr202/Tyr204), **b** phospho-Smad3 (Ser423/425), **c** phospho-Stat1(Tyr701) and **d** phospho-Stat3(Tyr705) was assessed by immunoblotting and normalized by total forms of Smad3, ERK1/2, Stat1, and Stat3. **e** Cardiac Fibroblasts were labeled with CFSE and stimulated with aldosterone (10$^{-7}$ mmol/L). Proliferation was accessed by flow cytometry after 96 h. Data are expressed as mean ± SEM and representative figures ($n = 7$/group). One-way ANOVA followed by Dunnett's multiple comparisons test were used for statistical analysis. *$P < 0.05$ vs WT veh; †$p < 0.05$ vs M7+/Δ veh; ‡$p < 0.05$ M7+/Δ aldo vs WT aldo; #$p < 0.05$ M7+/Δ salt vs WT salt; $^{\alpha}$ $p < 0.05$ M7+/Δ-Mg$^{2+}$ aldo vs M7+/Δ aldo; $^{\beta}p < 0.05$ M7+/Δ-Mg$^{2+}$ salt vs M7+/Δ salt; $^{\varepsilon}$ $p < 0.05$ M7+/Δ-Mg$^{2+}$ aldo-salt vs M7+/Δ aldo-salt.

Moreover, we found increased frequency of CD8 cytotoxic T cells, which induce tissue damage by the production of granzymes. These findings are in line with previous in vitro data in HEK cells expressing TRPM7 kinase-deficient mutants, which have increased aldosterone-induced MAPK activation, ICAM-1 expression, and ROS production[27].

Aldosterone induces activation of pro-fibrotic signaling pathways, including TGFβ, Smad, matrix metalloproteinase 2, lysyl oxidase, and MAPKs, which promote collagen deposition and cardiovascular fibrosis[2]. Corroborating this, aldosterone-salt caused significant cardiac, vascular and renal fibrosis in WT mice, effects that were amplified in TRPM7$^{+/\Delta kinase}$ mice. At the molecular level, exaggerated fibrosis in TRPM7-deficient mice was associated with increased phosphorylation of Smad3 and ERK1/2 compared with WT counterparts, whereas phosphorylation of Stat1 and expression of TGFβ, which were already

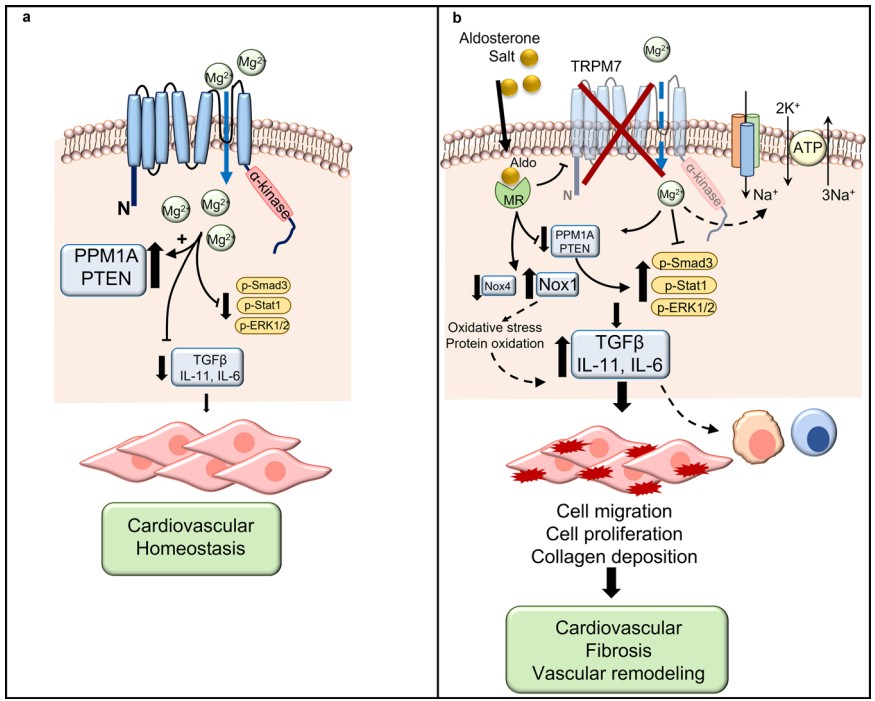

**Fig. 8 Interactions between TRPM7, Mg$^{2+}$, and aldosterone/salt. a** Under physiologic conditions functional TRPM7 regulates cellular Mg$^{2+}$ homeostasis and intracellular free Mg$^{2+}$ concentration ([Mg$^{2+}$]i), which influences signaling molecules, cell function, and cardiovascular homeostasis. **b** Aldosterone-salt stimulation causes downregulation of TRPM7, with associated decrease in [Mg$^{2+}$]i. This is associated with reduced activation of phosphatases PPM1A and PTEN; increased production of TGFβ, IL-11, and IL-6; increased phosphorylation of proinflammatory and profibrotic signaling molecules Smad3, Stat1, and ERK1/2; oxidative stress and potential imbalance of electrolytes (Na$^+$ and K$^+$). These mechanisms are involved in cell migration, proliferation, inflammation, and collagen deposition, resulting in vascular remodeling and cardiovascular fibrosis. Larger fonts = increased response; Smaller fonts = reduced responses. Arrows-activation; blocked line-inhibition; dashed line-pathways not yet proven.

increased in basal conditions in TRPM7$^{+/Δkinase}$ mice, were not further increased by treatments. These findings suggest that TRPM7-regulated pathways induced by aldosterone-salt are tightly controlled and not generalized phenomena. Potential upstream pathways contributing to this may involve Mg$^{2+}$-sensitive proteins including PPM1A and PTEN, which repress profibrotic signaling and which co-regulate each other[50]. PPM1A, a phosphatase that negatively controls TGFβ signaling by decreasing Smad2/3 activity, was downregulated in the heart, kidney, and aorta from aldosterone-salt-treated TRPM7-deficient mice, with similar patterns for PTEN. Tissues from non-treated TRPM7-deficient animals exhibit increased expression of PTEN. Mechanisms underlying these results are still elusive, but it is possible that some still unknown target protein of TRPM7-kinase might be a regulator or PTEN expression.

Another potential molecular player in cardiovascular fibrosis associated with hyperaldosteronism is Nox-associated oxidative stress[51]. Expression of Nox1, a major ROS-generating oxidase in cardiovascular tissue, was found to be increased in basal conditions from TRPM7$^{+/Δkinase}$ mice and in treated WT animals. On the other hand, Nox4, which has been shown to have cardiovascular protective functions through generation of H$_2$O$_2$[51], was significantly downregulated in treated TRPM7$^{+/Δkinase}$ mice versus WT counterparts. Hence potential cardiac protection by Nox4 was attenuated in TRPM7$^{+/Δkinase}$ mice. Nox2, which is typically expressed in phagocytic cells, was not influenced by treatment or TRPM7 status. Together our findings indicate downregulation of Nox4 and upregulation of Nox1 in TRPM7$^{+/Δkinase}$ mice, which may play a role in redox-regulated cardiovascular fibrosis.

Since TRPM7 deficiency causes perturbed cellular Mg$^{2+}$ homeostasis, it is possible that altered aldosterone-mediated responses in TRPM7$^{+/Δkinase}$ mice relate to intracellular Mg$^{2+}$

depletion. This was confirmed for some signaling pathways in cardiac fibroblasts, particularly activation of ERK1/2, Smad3, Stat1, TGFβ1, and IL-6 since Mg$^{2+}$ supplementation normalized aldosterone-salt effects in TRPM7-deficient fibroblasts. However, this was not evident for IL-11, highlighting the complexity of interplay between aldosterone, TRPM7, and Mg$^{2+}$. We also found that impaired cell growth in TRPM7$^{+/Δkinase}$ fibroblasts was ameliorated by Mg$^{2+}$ supplementation.

In summary, our data show that TRPM7 kinase deficiency, which is associated with hypomagnesemia and reduced intracellular Mg$^{2+}$ concentration, increases susceptibility to cardiovascular and renal fibrosis induced by aldosterone and salt. These phenomena are associated with marked natriuresis, possibly linked to altered renal sodium handling by downregulation of Na$^+$, K$^+$ATPase. Molecular mechanisms underlying cardiovascular and renal injury involve downregulation of PPM1A and PTEN and associated upregulation of Smad3 and ERK1/2, processes that are Mg$^{2+}$-sensitive. In conclusion, our study defines TRPM7 kinase-regulated pathways and highlights a protective role for TRPM7-Mg$^{2+}$ in cardiovascular and renal fibrosis induced by aldosterone. Targeting this system may have therapeutic potential in conditions associated with hyperaldosteronism.

## Methods
Please see Supplementary Methods for detailed methods

**Animals**. Animal experiments were approved by the University of Glasgow Animal Welfare and Ethics Review Board in accordance with the United Kingdom Animals Scientific Procedures Act 1986 (Licence No. 70/9021) and with ARRIVE Guidelines[52]. Wild type (WT) mice (C57BL/6J and SV129 mixed background) and mice heterozygous for the deletion of the TRPM7-kinase (TRPM7$^{+/Δkinase}$), generated by the gene-targeting vector technique[24]. Reverse transcription PCR analysis was used to identify wildtype (WT) (TRPM7+/+)

and heterozygous (TRPM7[+/Δkinase] (TRPM7+/Δ)) animals. Homozygous mice TRPM7[Δkinase/Δkinase] are embryonic lethal.

**Animal treatment**. WT and TRPM7[+/Δkinase] male mice, 12–16 weeks of age were studied. Surgical procedures were performed under anesthesia. WT and TRPM7[+/Δkinase] mice were divided into 4 groups and treated for 4 weeks: Group 1 - vehicle controls (veh, WT $n = 14$; TRPM7[+/Δkinase] $n = 17$); Group 2 - aldosterone-infused (aldosterone group, WT $n = 10$; TRPM7[+/Δkinase] $n = 9$) (600 µg/kg/day) by Alzet osmotic mini-pumps; Group 3 - 1% NaCl drinking water (salt group, WT $n = 7$; TRPM7[+/Δkinase] $n = 9$); Group 4 - aldosterone-infused+1% NaCl drinking water (aldosterone-salt, WT $n = 13$; TRPM7[+/Δkinase] $n = 15$). Mice were euthanized after 4 weeks of treatment. Aorta, mesenteric vascular bed, heart, kidneys, and spleens were dissected for further analysis.

**Plasma and urine biochemistry**. Blood was collected under isoflurane anesthesia by cardiac puncture immediately prior to sacrifice. Spot urine was collected from the bladder during sacrifice and snap frozen in liquid nitrogen. Concentrations of calcium, phosphate, sodium, potassium, chloride, magnesium, albumin, creatinine, plasma glucose were determined by an automated analyzer.

**Culture of cardiac fibroblasts**. Hearts from WT and TRPM7[+/Δkinase] mice were cut into small pieces and digested with collagenase and pancreatin. The final pellet was resuspended and plated. Fibroblasts were kept in culture and experiments performed (up to passage 8). Fibroblasts were maintained in 0.5% FBS for 24 h before experiments. Cells were treated with salt (NaCl 40 mM, Sigma-Aldrich, Dorset, UK) or aldosterone ($10^{-7}$ M) or salt plus aldosterone for varying time periods, and protein expression assessed by immunoblotting. Choline Chloride (40 mM, Sigma-Aldrich, Dorset, UK) was used as an osmotic control.

**Blood pressure measurement**. Systolic blood pressure (SBP) was measured by tail-cuff plethysmography (BP 2000 Blood Pressure Analysis System, Visitech, Science Products GmbH, Germany). Mice were immobilized on a warmed platform at 37 °C. One week before commencing the experiments, mice were trained daily using the Visitech system. By the time the study started, baseline blood pressures in all mice were stable. During the experimental period, blood pressure was measured in conscious mice twice weekly for 4 weeks. The first 5 recordings were disregarded to avoid artefact and the average of the 10 successive measurements were taken as the final blood pressure reading. This protocol has been well described and validated[53].

**Histology**. Hearts, aortas, and kidneys were fixed in 10% buffered-formalin solution and processed for histological inclusion in paraffin. Five-µm thick tissue sections were stained with PicroSirius red for light microscopy.

**Functional studies in mesenteric resistance arteries**. First- and second-order mesenteric resistance arteries were cut into segments and mounted on a wire myograph, as previously described[54]. Contractile responses were assessed by adding KCl and endothelial integrity was verified by relaxation induced by acetylcholine (Ach) in pre-contracted vessels. Cumulative concentration–response curves to phenylephrine (Phe) were obtained. Endothelium-dependent relaxation was assessed by concentration-responses to Ach in Phe-pre-contracted vessels. Endothelium-independent relaxation was assessed by concentration responses to sodium nitroprusside (SNP).

**Pressure myography**. Vascular structure and mechanics were assessed in resistance arteries prepared as pressurized systems as previously described[54]. Pressure–diameter curve obtained by progressively increasing intraluminal pressure. Internal and external diameters were used to calculate parameters such as: wall thickness, cross-sectional area (CSA), and wall:lumen ratio[55]. Mechanical properties were assessed by stress–strain curves previously described[54,55].

**Measurement of tissue Mg²⁺**. Dried tissues were weighed using an analytical balance, followed by digestion in nitric acid. $Mg^{2+}$ concentration was analyzed by colorimetric reaction using a commercial kit[56].

**Flow cytometry**. Cells from kidneys and spleens were resuspended in FACS buffer, blocked with normal rat serum 5% in PBS, and stained with fluorescent-conjugated anti-mouse monoclonal antibodies: anti-CD45-FITC, anti-CD3-PE-Cy7, anti-CD4-APC, anti-CD8-APC-Cy7, anti-F4/80-Alexa-647, anti-CD11c-PE-Cy7, anti-CD206-FITC, and anti-CD45-PE. Data acquisition was performed by flow cytometry.

**Real-time reverse-transcription polymerase chain reaction (PCR)**. Total RNA was isolated. cDNA was generated from total RNA and real-time PCR reaction performed. Specific murine primers to GAPDH, TRPM7, TRPM6, fibronectin, and collagen-1 were used (Supplementary Table 1). Relative gene expression was calculated by the $2^{-\Delta\Delta Ct}$ cycle threshold method as previously described[57].

**Immunoblotting**. Proteins from hearts and kidneys were separated by electrophoresis on a polyacrylamide gel and transferred onto a nitrocellulose membrane. Non-specific binding sites were blocked with 5% non-fat dry milk in TBS-T. Membranes were incubated with the following primary specific antibodies: β-actin, phospho-Smad3, peroxiredoxin-SO3 (Prs-SO₃H), α-tubulin, IL-6, TGFβ1; phospho-Stat3 (Tyr705), total-Stat3, phospho-Stat1 (Tyr701), total-Stat1, phospho-ERK1/2 (Thr202/Tyr204), total-Smad3, total-ERK1/2, PTEN, IL-11, ox-PTP, PPM1A, Nox4, Nox1 Nox2, TRPM7 and phospho-TRPM7 (Ser1511)[29]. Membranes were washed and incubated with secondary fluorescence-coupled antibodies goat-anti-mouse-IRDye 680 or goat-anti-rabbit-IRDye 800 and visualized by an infrared laser scanner. Protein expression levels were normalized to loading controls.

**Hydrogen peroxide production**. Hydrogen peroxide was evaluated in heart tissues using the Amplex Red Hydrogen Peroxide/Peroxidase Assay Kit (Molecular Probes/Life Technologies, Paisley, United Kingdom) according to the manufacturer´s instructions. Obtained values were normalized by protein concentration in the tissue lysate.

**Proliferation assay**. The cell tracking dye carboxyfluorescein succinimidyl ester (CFSE) was used to assess cardiac fibroblast proliferation.

**Statisticals and reproducibility**. Data are presented as mean ± SEM. Two-tailed unpaired Student's t-test was used when differences between two groups were analyzed. Analysis of variance (ANOVA) and the Dunnett's multiple comparisons test were used to evaluate statistical significance of differences between three or more groups. For vascular function, concentration-response data were analyzed by determining EC50 and maximal response (Emax) values from experimental data fitted to a four-parameter logistic function against the null hypothesis. Differences in vascular structure were assessed using values of Area Under the Curve (AUC). Differences in vascular mechanics were assessed using values of slope of stress–strain curves. Data analysis was conducted using GraphPad Prism6.0. Significance was assumed if $p < 0.05$.

**Reporting summary**. Further information on research design is available in the Nature Research Reporting Summary linked to this article.

## Data availability

The data that support the findings of this study are available from the corresponding authors on reasonable request. Uncropped and unedited blot/gel images are provided in the Supplementary Figs. 20–25. All data underlying the graphs and charts presented in the main figures are provided as Supplementary Data in Excel format.

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

## Acknowledgements

R.M.T. is supported by grants from the British Heart Foundation (BHF) (CH/12/4/29762; RE/18/6/34217). A.C.M. is supported by a Walton fellowship, University of Glasgow and Z.-G.Z. by a China Scholarship Council grant (201708060309). V.C. and T.G. were supported by the Deutsche Forschungsgemeinschaft, Transregional Collaborative Research Center 152 and Research Training Group 2338.

## Author contributions

R.M.T. designed the study and provided funding, critical discussion, and final preparation and submission of the manuscript; F.J.R. designed the study, performed experiments, analyzed data, prepared the figures, and wrote the manuscript draft; Z.G.Z., A.P.H., K.Y.H., L.L.C., K.B.N., S.E.F.N., R.A.-L., A.C., and M.Z. performed experiments. A.C.M. designed the study, performed experiments, and provided critical discussion. A.G.R. and L.R. provided the mice and critical discussion. V.C. and T.G. provided anti-pTRPM7 antibody and critical discussion.

## Competing interests

The authors declare no competing interests.
