## [Peer Review File · Communications Biology]

Reviewers' comments:

Reviewer #1 (Remarks to the Author):

In this study, the authors describe the phenotype of M7/+ Δ mice, which have a mutation in the kinase domain of TRPM7, at baseline and in response to aldosterone, salt feeding or both. These animals seem to have augmented hypertension, impaired endothelium-dependent vascular relaxation and enhanced fibrosis, among other findings. While this is interesting, the paper largely comes across as a cataloguing of findings without establishing a fundamental cause of these abnormalities. Given that increased blood pressure might account for many of these findings, it would be interesting to normalize the blood pressure in the M7/+ Δ mice to see if this prevents the vascular and renal abnormalities. There are a variety of specific issues that should also be addressed:

In figure 1, you should show the effects of aldo on WT and your kinase ko mice in the same plot and show us statistical analysis comparing these two. A better way to show this would be to show the effect of aldo only on WT and kinase deficient mice in one plot, effect of salt in another, and effect of salt + aldo in yet another.

You have made these comparisons in supplemental table 2, but it's a shame to relegate these important comparisons to the supplemental section. Also in suppl table 2, isn't there a difference between kinase deficient and WT systolic BPs with aldo and salt? A symbol for this (\pm) is lacking.

Did you use live/dead staining in your flow cytometry? This is important because dead or dying cells can nonspecifically take up dyes.

When you say non-specific binding sites were blocked, you should say how.

Page 8: The sentence: "Plasma magnesium levels were significantly reduced in TRPM7/+ Δ kinase mice compared with WT controls (Table 2)", should read: "As baseline, plasma magnesium levels were significantly reduced in TRPM7/+ Δ kinase mice compared with WT controls (Table 2)."

It is a shame that telemetry wasn't used to measure blood pressure in these animals.

In the example Western in figure 2, the phosphorylation of TRPM7 in the M7/+ Δ mice is increased markedly by salt, but this is not reflected by the bar graph.

On the first line of page 14, you say that TRPM7 activity is decreased by aldo-salt. Is this entirely based on its phosphorylation? In the referenced work it seems this is due to autophosphorylation. If the kinase dead enzyme really has an increase in phosphorylation in response to salt (see comment above), this wouldn't seem to be due to autophosphorylation. What upstream kinase might be involved.

The most striking finding in table 2 is that the urinary sodium in response to salt and aldo is increased more than 4-fold (from 170 to 710) in the M7/+ Δ mice. There are two issues here. First, without known sodium intake, this finding is impossible to interpret, but if these animals are at steady state, it indicates that they are consuming much more sodium than WT mice. The reason for this is unclear, but perhaps there is a central effect of the kinase portion of the enzyme.

Reviewer #2 (Remarks to the Author):

In this manuscript, the authors demonstrated that TRPM7 deficiency underlined the hypertension status and cardiovascular damage of hyperaldosteronism. However, there are several questions to be addressed.

1. I noticed that a similar paper was published in 2008 (DOI: 10.1161/HYPERTENSIONAHA.107.100339). The authors may further discuss the difference and improvement of the current study.

2. Figure 1 showed the SBPs of WT and TRPM7+/ Δ kinase mice in A and B, respectively. The comparison of SBPs between aldosterone-salt induced WT and TRPM7+/ Δ kinase mice could not be found in the related part, thus the effect of TRPM7 on blood pressure could not be drawn from the study in animal model.
3. Figure 2 only shows that the level of p-TRPM7(Ser1511) standardized by α -tubulin to state that the activation of TRPM7. As "TRPM7 gene expression was increased by salt and reduced in the aldosterone-salt group in WT mice", it might be better to also present the total expression of TRPM7 and show the ratio of p-TRPM7(Ser1511) : total TRPM7.
4. The combined use of aldosterone and salt exaggerated the cardiovascular and renal fibrosis than the single use of aldosterone or salt in animal model. But authors only used aldosterone to stimulate cardiac fibroblast in vitro, which was different from in vivo models. Also, only testing the level of Nox4, H₂O₂, Prx-SO₃ and PTP in heart might not fully support the conclusion that "aldosterone-salt-induced oxidative stress in the heart is independent of TRPM7". It would be better to add both aldosterone and salt to the fibroblasts from TRPM7+/ Δ kinase mice and WT counterparts in vitro.
5. The study mainly demonstrated that TRPM7 deficiency played an important role in hypertension as well as cardiovascular and renal fibrosis induced by aldosterone-salt. This effect was Mg²⁺-regulated. As known, TRPM7 can also release TRPM7 cleaved kinases, which will transfer into the nucleus and regulate the gene expression. Would TRPM7 cleaved kinases also play roles in the aldosterone-salt induce target organ damage?
6. Please add a schematic showing the whole study.

Response to reviewer 1 comments

We would like to thank the reviewer for his/her constructive comments. The paper has undergone major revision to include data from new experiments as suggested by the reviewers.

We would like to highlight that while we have made every attempt to respond to each comment raised by the reviewers, there were some experiments that we could not do due to challenges and restrictions posed by the COVID19 situation. In particular, due to university closure and associated issues, we have been unable to undertake new *in vivo* studies due to lack of availability of transgenic mice. We had to cull our mouse colonies during the lockdown and are only now managing to rederive our transgenic strains. Nevertheless, we performed new *ex vivo* and *in vitro* studies that address, at least in part, some of the reviewers' comments. We truly hope the editor and reviewers take these challenges into consideration when reassessing our paper.

All changes made are detailed below. We believe that the paper is significantly improved and hope that it is now acceptable for publication. Our responses to each comment are highlighted in blue.

Reviewer #1 (Remarks to the Author):

In this study, the authors describe the phenotype of M7+/ Δ mice, which have a mutation in the kinase domain of TRPM7, at baseline and in response to aldosterone, salt feeding or both. These animals seem to have augmented hypertension, impaired endothelium-dependent vascular relaxation and enhanced fibrosis, among other findings. While this is interesting, the paper largely comes across as a cataloguing of findings without establishing a fundamental cause of these abnormalities. Given that increased blood pressure might account for many of these findings, it would be interesting to normalize the blood pressure in the M7+/ Δ mice to see if this prevents the vascular and renal abnormalities.

The reviewer raises an important point. To dissect whether cardiovascular and renal changes are dependent or independent of augmented blood pressure responses in M7+/ Δ mice, we would have liked to perform new *in vivo* studies treating mice with agents that reduce blood pressure and/or normalise magnesium homeostasis. Unfortunately we have not been able to perform new animal experiments due to numerous challenges related to the Covid19 pandemic. Specifically, our labs were closed for many months and we needed to cull our mouse colonies. Since (partial) reopening of our labs, we have been attempting to re-derive our mouse colonies, but due to the M7+/ Δ mice being 'slow breeders', it is taking a long time to repopulate our colonies. Considering that we would need a large number of mice to perform the new studies as suggested by the reviewer (WT and M7+/ Δ mice treated with vehicle, aldosterone, salt, aldosterone+salt in the absence and presence of antihypertensive drugs and/or magnesium supplementation) we anticipate these studies will only be completed in 12-14 months time. Accordingly, while we would have liked to perform the new *in vivo* studies suggested by the reviewer, we are not able to do so at this time. Hopefully the reviewer will appreciate and understand these challenges. Nevertheless, we have attempted to answer the concerns raised using *ex vivo* and cell-based models.

Using an *in vitro* strategy, we performed new experiments to evaluate whether some of the changes observed *in vivo* can be ameliorated by magnesium. We showed that TRPM7-

deficient animals exhibit increased sensitivity to blood pressure and cardiovascular fibrosis induced by aldosterone and salt. Because these animals exhibit hypomagnesemia, an important question to address is whether the pro-fibrotic effects are dependent on Mg^{2+} , linked to TRPM7 deficiency. We sought to address this question using *in vitro* approaches.

Hearts from TRPM7^{+/Δ} mice treated with aldosterone and salt exhibit increased phosphorylation of ERK, Smad3, and Stat1 and expression of TGFβ1 and IL-6 (**revised manuscript, Figure 7 and supplementary figure S5**). Similar results were observed in primary culture of cardiac fibroblasts treated with aldosterone and salt. These effects were ameliorated by Mg^{2+} treatment, suggesting that at least part of these phenomena are Mg^{2+} -dependent (**revised manuscript, Figure 8 and supplementary figure S7**).

In figure 1, you should show the effects of aldo on WT and your kinase ko mice in the same plot and show us statistical analysis comparing these two. A better way to show this would be to show the effect of aldo only on WT and kinase deficient mice in one plot, effect of salt in another, and effect of salt + aldo in yet another. You have made these comparisons in supplemental table 2, but it's a shame to relegate these important comparisons to the supplemental section. Also in suppl table 2, isn't there a difference between kinase deficient and WT systolic BPs with aldo and salt? A symbol for this (‡) is lacking.

We thank the reviewer for the suggestion. We have reorganised the presentation of the graphs (**revised manuscript figure 1A-C**) and agree the data are easier to understand.

The reviewer's observation regarding differences in blood pressure between aldo and aldo+salt M7^{+/Δ} mice is correct. We have added the symbol of significance, particularly at week 3 (‡) (**revised manuscript, figure 1C**).

Did you use live/dead staining in your flow cytometry? This is important because dead or dying cells can nonspecifically take up dyes. When you say non-specific binding sites were blocked, you should say how.

We thank the reviewer for highlighting this important point. In our experiments we did not use live/dead staining. However, to minimise differences in cell isolation between the groups, we always performed experiments using WT and TRPM7^{+/Δ} animals at the same time. For blocking the unspecific binding, we used normal rat serum 5% v/v in PBS. This information has been added in the current version of the article (**revised manuscript, page 7, line 7; Supplementary text page 5; topic Flow cytometry, line 2-3**)

Page 8: The sentence: "Plasma magnesium levels were significantly reduced in TRPM7^{+/Δ}kinase mice compared with WT controls (Table 2)", should read: "As baseline, plasma magnesium levels were significantly reduced in TRPM7^{+/Δ}kinase mice compared with WT controls (Table 2)."

We would like to thank the reviewer for his/her suggestion. We have made the appropriate changes in the main text (**revised manuscript, page 9, paragraph 2, line 1-2**)

It is a shame that telemetry wasn't used to measure blood pressure in these animals.

We totally agree that telemetry is the gold standard for blood pressure measurement. However, this study was performed with many groups of animals, that would require numerous probes and platforms. This was not practical or feasible. In the present study we used

plethysmography, a well established and optimized technique. To minimize errors and variation in blood pressure measurements, all animals were trained to the apparatus a week before experimentation. By day zero of the study, blood pressures were stabilized in each animal. During the experimental period, blood pressure was measured twice weekly by the same person. To clarify this, we have provided additional information in the revised manuscript (**revised manuscript, page 6, topic Blood pressure measurement**).

In the example Western in figure 2, the phosphorylation of TRPM7 in the M7+/ Δ mice is increased markedly by salt, but this is not reflected by the bar graph.

We agree with the reviewer that the images were not good representatives of the data. To address this issue, we extracted fresh renal protein lysates from all animals used in the different groups and performed new western blot studies. As a positive and negative control for the antibody, we used tissues from WT and TRPM7 kinase dead mice (TRPM7^{R/R}) that contain a K1646R point mutation and exhibit deficiency of the catalytic activity of TRPM7 kinase domain in the whole organism (Kaitsuka T, et al. Sci Rep. 2014 Jul 17;4:5718. Ryazanova LV et al. Scientific reports. 2014 Dec 23;4:7599) (**revised manuscript, figure 2A, 2B, new data**).

On the first line of page 14, you say that TRPM7 activity is decreased by aldo-salt. Is this entirely based on its phosphorylation? In the referenced work it seems this is due to autophosphorylation. If the kinase dead enzyme really has an increase in phosphorylation in response to salt (see comment above), this wouldn't seem to be due to autophosphorylation. What upstream kinase might be involved.

We thank the reviewer for the observation and suggestions. TRPM7 contains a cytosolic α -kinase domain with an Mg²⁺/ATP-binding site, which is critical for the catalytic activity, and a point mutation on the lysine 1646 abolishes the kinase activity. Autophosphorylation of the TRPM7 kinase domain on serine 1511/1567 enhances kinase–substrate interactions leading to their serine/threonine phosphorylation (Matsushita M et al. J Biol Chem. 2005 27;280:20793-803). Animals that contain a K1646R point mutation exhibit deficiency of the catalytic activity of TRPM7 kinase domain in the whole organism and show reduced phosphorylation of the Ser1511 as demonstrated in this work (**revised manuscript, figure 2A, new data**) and by others (Nat Commun. 2017 Dec 4;8(1):1917). We agree with the reviewer that the autophosphorylation may not be the only way to activate the kinase domain and that TRPM7 may also be a target for other serine/threonine kinases, that are still unknown. However, in our results we also found reduced TRPM7 total protein expression (**revised manuscript, figure 2B; Discussion, page 15, first paragraph**) in WT treated with aldosterone-salt and in TRPM7+/ Δ mice. These new experiments were performed using an antibody specific to the residues 1146-1165, that are upstream to the truncated residue in TRPM7+/ Δ mice (1538-1863). These new data show that WT animals treated with aldosterone-salt and TRPM7+/ Δ mice exhibit both reduced TRPM7 channel expression and kinase phosphorylation.

The most striking finding in table 2 is that the urinary sodium in response to salt and aldo is increased more than 4-fold (from 170 to 710) in the M7+/ Δ mice. There are two issues here. First, without known sodium intake, this finding is impossible to interpret, but if these animals are at steady state, it indicates that they are consuming much more sodium than WT mice. The reason for this is unclear, but perhaps there is a central effect of the kinase portion of the enzyme.

We agree with the reviewer that the increased concentration of urinary sodium in TRPM7^{+/Δ} mice treated with aldosterone-salt is a very interesting finding. We did not perform metabolic studies and accordingly can not comment on the exact salt intake for each animal. As the reviewer points out, our thesis is that the kinase domain of TRPM7^{+/Δ} influences sodium metabolism, a possibility that we have addressed with new experiments.

To address the potential role of TRPM7 kinase on molecular processes that regulate sodium homeostasis, we focused on expression of major sodium transporters in the kidney, specifically ENaC and Na,K-ATPase. These are important regulators of renal sodium reabsorption and are modulated by aldosterone.

We found increased expression of ENaC induced by aldosterone-salt in both groups. However, Na,K-ATPase was increased only in WT mice treated with aldosterone-salt. Na,K-ATPase is a recognised Mg²⁺ dependent enzyme expressed in the basolateral surface of the nephron. In physiologic concentrations of Mg²⁺, Na,K-ATPase pumps Na⁺ from the cytoplasm and takes up K⁺. Reduced intracellular [Na⁺] will force the Na⁺ gradient through the luminal surface, therefore increasing the Na⁺ reabsorption (Mayan H, et al., *Physiol Rep*, 2018;6:e13843). These effects may be attenuated in conditions where Na,K-ATPase is downregulated as observed in TRPM7-deficient/hypomagnesemic mice. Mechanisms underlying these results are still unclear but might be dependent on reduced Mg²⁺ levels, since Na,K-ATPase (in fact all ATPases) are Mg²⁺-dependent enzymes. The new data have been added in **Supplemental Figure 1A, 1B and in the discussion (revised manuscript, pages 15-16)**.

Findings from these studies suggest for the first time that renal sodium handling, is regulated by TRPM7-dependent mechanisms, possibly via altered sodium transport. This interesting paradigm requires further examination, with studies planned in the future as our mouse colonies grow.

Response to reviewer 2 comments

We would like to thank the reviewer for his/her constructive comments. The paper has undergone major revision to include data from new experiments as suggested by the reviewers.

We would like to highlight that while we have made every attempt to respond to each comment raised by the reviewers, there were some experiments that we could not do due to challenges and restrictions posed by the COVID19 situation. In particular, due to university closure and associated issues, we have been unable to undertake new *in vivo* studies due to lack of availability of transgenic mice. We had to cull our mouse colonies during the lockdown and are only now managing to redrive our transgenic strains. Nevertheless, we performed new *ex vivo* and *in vitro* studies that address, at least in part, some of the reviewers' comments. We truly hope the editor and reviewers take these challenges into consideration when reassessing our paper.

All changes made are detailed below. We believe that the paper is significantly improved and hope that it is now acceptable for publication. Our responses to each comment are highlighted in blue.

Reviewer #2 (Remarks to the Author):

In this manuscript, the authors demonstrated that TRPM7 deficiency underlined the hypertension status and cardiovascular damage of hyperaldosteronism. However, there are several questions to be addressed.

1. I noticed that a similar paper was published in 2008 (DOI: 10.1161/HYPERTENSIONAHA.107.100339). The authors may further discuss the difference and improvement of the current study.

We thank the reviewer for highlighting our previous publication. In our 2008 paper we showed that aldosterone-treated C57Bl/6 mice exhibited increased cardiovascular fibrosis and reduced TRPM7 expression through Mg^{2+} dependent mechanisms. These findings highlighted an important relationship between aldosterone, Mg^{2+} and cardiovascular fibrosis in hypertension and suggested that TRPM7 may play a role. The 2008 paper formed the foundation and background for the present study, which focuses specifically on the putative role of TRPM7 kinase in aldosterone- and salt- sensitive hypertension and potential Mg^{2+} -dependent mechanisms. To address this we used a novel mouse model, TRPM7^{+/ Δ} mice, which have a point mutation in the kinase domain. The importance of TRPM7 kinase is evidenced by the fact that these mice are hypomagnesemic. To explore whether aldosterone/salt-induced cardiovascular fibrosis in TRPM7-deficient mice involves a Mg^{2+} -dependent process, we have performed new studies where primary culture of cardiac fibroblasts from TRPM7^{+/ Δ} mice were stimulated with aldosterone and salt. We explored molecular processes that underlie fibrosis, inflammation and cell growth by studying phosphorylation of ERK, Smad3, and Stat1, expression of TGF β 1 and IL-6 and cell proliferation (**revised manuscript, figure 8, new data. supplementary figure S7, new data**). Similar to what we observed in cardiac tissue from *in vivo* studies in aldosterone/salt-treated TRPM7^{+/ Δ} mice, profibrotic and proinflammatory signaling was upregulated by aldosterone/salt, with cell growth inhibition. These responses were augmented in TRPM7^{+/ Δ} cardiac fibroblasts versus control wildtype counterparts. The

aberrant effects in TRPM7-deficient cells were ameliorated by Mg^{2+} treatment, suggesting that, at least in part, these phenomena are promoted by Mg^{2+} deficiency. Implications of these findings have been added in the discussion section (**revised manuscript, page 17, paragraph 2, line 1-3.**

2. Figure 1 showed the SBPs of WT and TRPM7^{+/-}Δkinase mice in A and B, respectively. The comparison of SBPs between aldosterone-salt induced WT and TRPM7^{+/-}Δkinase mice could not be found in the related part, thus the effect of TRPM7 on blood pressure could not be drawn from the study in animal model.

We thank the reviewer for the suggestion. To better demonstrate differences between groups with respect to treatments, we have reorganised the graphs as shown in figure 1 of the revised manuscript. This allows the comparisons between WT and TRPM7^{+/-}Δkinase mice according to treatments: **Figure 1A – WT and TRPM7^{+/-}Δkinase vehicle and treated with aldosterone; Figure 1B WT and TRPM7^{+/-}Δkinase vehicle and treated with salt; Figure 1C - WT and TRPM7^{+/-}Δkinase vehicle and treated with aldosterone-salt. TRPM7^{+/-}Δkinase mice exhibit increased blood pressure vs WT after aldosterone or salt treatment.**

3. Figure 2 only shows that the level of p-TRPM7(Ser1511) standardized by α-tubulin to state that the activation of TRPM7. As “TRPM7 gene expression was increased by salt and reduced in the aldosterone-salt group in WT mice”, it might be better to also present the total expression of TRPM7 and show the ratio of p-TRPM7(Ser1511) : total TRPM7.

To address this question, we extracted fresh renal protein lysates from all animals used in the different groups and performed new studies by western blotting. As positive and negative controls for the antibody, we used tissues from WT mice and TRPM7 kinase dead mice (TRPM7R/R) that contains a K1646R point mutation and exhibits deficiency of the catalytic activity of TRPM7 kinase domain in the whole organism (Kaitsuka T, et al. Sci Rep. 2014 Jul 17;4:5718. Ryazanova LV et al. Scientific reports. 2014 Dec 23;4:7599). Reduced p-TRPM7 (Ser1511) was found in kidneys from WT mice treated with aldosterone- salt and in tissues from TRPM7^{+/-}Δkinase mice. Additionally, using an antibody specific to the residues 1146-1165, which are upstream to the truncated residue in TRPM7^{+/-}Δkinase mice (1538-1863), we found reduced TRPM7 expression in tissues from WT mice treated with aldosterone-salt and in TRPM7^{+/-}Δ mice. The new data are added in **figure 2A, 2B (revised manuscript).**

We agree with the reviewer that the best way to represent the data would be by normalising phospho-TRPM7 by total-TRPM7. However, since both antibodies are from rabbit, it is not appropriate to normalise phospho-TRPM7 by total TRPM7 under these conditions.

4. The combined use of aldosterone and salt exaggerated the cardiovascular and renal fibrosis than the single use of aldosterone or salt in animal model. But authors only used aldosterone to stimulate cardiac fibroblast in vitro, which was different from in vivo models.

We thank the reviewer for the suggestion. To address these concerns, we performed new studies using primary culture of cardiac fibroblasts from WT, TRPM7^{+/-}Δkinase and TRPM7^{+/-}Δkinase plus Mg^{2+} . Cells were treated with NaCl (40 mM), aldosterone (10^{-7} M) or aldosterone (10^{-7} M) plus NaCl (40mM). Choline Chloride (40 mM) was used as osmotic control. We investigated the activation of Smad3, Stat1, ERK1/2 after 10 min stimulation and the expression of IL-6, IL-11 and TGFβ1 after 24 h stimulation. The new data are shown in **figure 8 and supplemental figure 7 of the revised manuscript.**

Also, only testing the level of Nox4, H₂O₂, Prx-SO₃ and PTP in heart might not fully support the conclusion that “aldosterone-salt-induced oxidative stress in the heart is independent of TRPM7”. It would be better to add both aldosterone and salt to the fibroblasts from TRPM7^{+/Δkinase} mice and WT counterparts in vitro.

We thank the reviewer for this suggestion. To further explore the potential role of reactive oxygen species and oxidative stress in our model, we have performed additional studies, as suggested.

We have now performed additional studies exploring expression of Nox1 and Nox2. Hearts from WT mice exhibit increased expression of Nox1 by aldosterone and salt treatments. However, hearts from TRPM7^{+/Δkinase} mice exhibit increased Nox1 at baseline, which was not further increased after treatments. No changes in Nox2 expression were found.

Based on the new data we have modified our conclusions indicating that Nox isoforms are differentially expressed in TRPM7-deficient mice. Since Nox1 is a major ROS-generating oxidase in the cardiovascular system, it may play a role in increased redox-dependent proinflammatory signaling in TRPM7-deficient/hypomagnesemic states. We also found that Nox4, which is cardioprotective, was downregulated in treated TRPM7^{+/Δkinase} mice, suggesting attenuation of Nox4-induced cardioprotection.

The new data and discussion have been added in **supplementary figures 6B, 6C and discussion section of the revised manuscript, discussion, page 18, paragraph 2, lines 2-4.**

Implications of differentially regulated Noxs and the role of oxidative stress in cardiovascular damage in TRPM7-deficient mice is still unclear and warrants further investigation. Our future studies will address this.

5. The study mainly demonstrated that TRPM7 deficiency played an important role in hypertension as well as cardiovascular and renal fibrosis induced by aldosterone-salt. This effect was Mg²⁺-regulated. As known, TRPM7 can also release TRPM7 cleaved kinases, which will transfer into the nucleus and regulate the gene expression. Would TRPM7 cleaved kinases also play roles in the aldosterone-salt induce target organ damage?

We thank the reviewer for this very interesting question. Indeed, it has been demonstrated that the C-terminal kinase domain of both TRPM7 and TRPM6 can be cleaved, translocate to the nucleus and induce epigenetic modifications, in a cell-type specific manner [Desai, B. N. et al. Dev Cell 22, 1149-1162, (2012). Krapivinsky G, et al Proc Natl Acad Sci U S A. 2017 22;114:E7092-E7100]. Therefore, it is likely that these phenomena also occur in our model of cardiovascular damage induced by aldosterone and salt. We have added a new commentary in the discussion addressing these issues (**revised manuscript, page 17, paragraph 1, lines 5-8**)

To investigate this we would need more specific antibodies and the fragments would need to be investigated by mass-spectrometry. These experiments were beyond the scope of our study, but certainly would be of interest for future studies.

6. Please add a schematic showing the whole study.

We thank the reviewer for the suggestion. We have now added a graphic abstract (**revised manuscript, figure 9**)

Reviewers' comments:

Reviewer #1 (Remarks to the Author):

Please see attached.

Reviewer #2 (Remarks to the Author):

It is a pleasure to receive the revised manuscript again. Thanks to the author who has made significant revisions to this article. However, there are several questions to be addressed.

Major comments:

1. In the measurement of vascular function and structure, TRPM7+/Δ appears to exhibit different effects than aldosterone-salt. For example, TRPM7+/Δ kinase mice exhibited increased sensitivity to ACh-induced relaxation, whereas maximal ACh-induced relaxation was significantly reduced in WT and TRPM7+/Δ kinase mice treated with aldosterone and aldosterone-salt. TRPM7+/Δ decreased vascular cross-sectional area and wall to lumen ratio, while aldosterone salt treatment increased wall to lumen ratio in WT mice. The authors may need to explain further how to understand the inconsistency in results.
2. Line 211-213, "we questioned...in both groups". However, this does not prove the effect of TRPM7 on sodium transporters. Unless the reverse experiment is supplemented.
3. In lines 267-269, it said treatments enhanced CD11c/CD206 expression. In contrast, Figure 3 showed no significant difference between the salt-treated group and the control group.
4. In lines 278-279, the authors mentioned that collagen deposition in hearts, kidneys and aortas was significantly increased in aldosterone-salt-treated WT mice versus vehicle-treated counterparts. However, as seen from figure 5F, collagen deposition in the aorta did not increase significantly.
5. Figure 9 is so simple that the connections between many signal molecules are not mapped. It is suggested that the authors should further improve it.

Minor:

1. The effect of Nox4 on cardiac remodeling is still controversial, and the author mentioned two inconsistent views of Nox4 in line 301 and line 435 respectively. It is suggested to unify or further discuss the viewpoints in this paper.
2. In Supplementary Figure S6A and S6B, the expression, upregulation of Nox1 and downregulation of Nox4, does not match with the figures in Line440-441.
3. There are some writing mistakes and grammatical errors.
 - (1) Line37, "ERK1/2 and Stat1 was upregulated": "was" might be corrected as "were".
 - (2) Line63-65, what is the meaning of the "Of these TRPM7"? It is suggested to make sure the sentence structure is accurate.
 - (3) Line219, "(Figure 1C 1A)" might be corrected as "(Figure 1C)".

Reviewer #1 (Remarks to the Author):

In this study, the authors describe the phenotype of M7+/Δ mice, which have a mutation in the kinase domain of TRPM7, at baseline and in response to aldosterone, salt feeding or both. These animals seem to have augmented hypertension, impaired endothelium-dependent vascular relaxation and enhanced fibrosis, among other findings. While this is interesting, the paper largely comes across as a cataloguing of findings without establishing a fundamental cause of these abnormalities. Given that increased blood pressure might account for many of these findings, it would be interesting to normalize the blood pressure in the M7+/Δ mice to see if this prevents the vascular and renal abnormalities.

The reviewer raises an important point. To dissect whether cardiovascular and renal changes are dependent or independent of augmented blood pressure responses in M7+/Δ mice, we would have liked to perform new in vivo studies treating mice with agents that reduce blood pressure and/or normalise magnesium homeostasis. Unfortunately we have not been able to perform new animal experiments due to numerous challenges related to the Covid19 pandemic. Specifically, our labs were closed for many months and we needed to cull our mouse colonies. Since (partial) reopening of our labs, we have been attempting to re-derive our mouse colonies, but due to the M7+/Δ mice being 'slow breeders', it is taking a long time to repopulate our colonies. Considering that we would need a large number of mice to perform the new studies as suggested by the reviewer (WT and M7+/Δ mice treated with vehicle, aldosterone, salt, aldosterone+salt in the absence and presence of antihypertensive drugs and/or magnesium supplementation) we anticipate these studies will only be completed in 12-14 months time. Accordingly, while we would have liked to perform the new in vivo studies suggested by the reviewer, we are not able to do so at this time. Hopefully the reviewer will appreciate and understand these challenges. Nevertheless, we have attempted to answer the concerns raised using ex vivo and cell-based models.

Using an in vitro strategy, we performed new experiments to evaluate whether some of the changes observed in vivo can be ameliorated by magnesium. We showed that TRPM7-deficient animals exhibit increased sensitivity to blood pressure and cardiovascular fibrosis induced by aldosterone and salt. Because these animals exhibit hypomagnesemia, an important question to address is whether the pro-fibrotic effects are dependent on Mg²⁺, linked to TRPM7 deficiency. We sought to address this question using in vitro approaches.

Hearts from TRPM7+/Δ mice treated with aldosterone and salt exhibit increased phosphorylation of ERK, Smad3, and Stat1 and expression of TGFβ1 and IL-6 (revised manuscript, Figure 7 and supplementary figure S5). Similar results were observed in primary culture of cardiac fibroblasts treated with aldosterone and salt. These effects were ameliorated by Mg²⁺ treatment, suggesting that at least part of these phenomena are Mg²⁺-dependent (revised manuscript, Figure 8 and supplementary figure S7).

At issue here is that the authors have framed the paper starting from the issue of hypertension. Given that they have done that, and it's in Fig. 1., it would seem reasonable of Reviewer #1 to request these additional studies. However, when I read the paper, I don't even think it's about hypertension, the framing is wrong. Since M7+aldo has high BP but not much fibrosis in the heart (fig 5), it would seem to suggest to me that the high BP alone is not driving the pathology. I think the authors approach and explanations and new in vitro experiments are ok for this point.

But the framing of the paper with Fig 1 starting with hypertension seems doesn't sit properly. If it's not that important, then it seems strange to start with it as Fig 1.

Recommendations:

I would suggest that:

- Fig 1 be downplayed as just a part of another figure.
- The statement in the abstract “aldosterone-mediated hypertension” be removed and read instead something like... “Our model revealed vascular dysfunction and cardiovascular-renal fibrosis were exaggerated in the TRPM7 het mice given aldosterone and salt.” Similar changes should be made throughout the results and discussion.
- A supplementary picture should be added to show more frames like in Fig 5, especially the cardiac fibrosis, as one frame with bad fibrosis could be selective. How many hearts were actually processed for PicroSirius red? There appear to be 8 different samples in which collagen is measured for M7het aldo salt. Were there 8 hearts? Or did they just measure the same heart repeatedly. Given how important this image is to the study, it is imperative that they show multiple hearts and multiple areas of those hearts.

In figure 1, you should show the effects of aldo on WT and your kinase ko mice in the same plot and show us statistical analysis comparing these two. A better way to show this would be to show the effect of aldo only on WT and kinase deficient mice in one plot, effect of salt in another, and effect of salt + aldo in yet another. You have made these comparisons in supplemental table 2, but it's a shame to relegate these important comparisons to the supplemental section. Also in suppl table 2, isn't there a difference between kinase deficient and WT systolic BPs with aldo and salt? A symbol for this (‡) is lacking.

We thank the reviewer for the suggestion. We have reorganised the presentation of the graphs (revised manuscript figure 1A-C) and agree the data are easier to understand.

The reviewer's observation regarding differences in blood pressure between aldo and aldo+salt M7+/Δ mice is correct. We have added the symbol of significance, particularly at week 3 (‡) (revised manuscript, figure 1C).

There are many issues with Fig. 1. in addition to the question of hypertension driving pathology, and the lack of telemetry data. Fig 1B is confusing (no difference except in 1 value at 1 timepoint – is this an anomaly? If not, what does it mean, if anything?). And Fig 1C is also confusing. What is happening at Week 3 here – why is one of the red lines going up temporarily and another one going down temporarily. What physiologic explanation would reasonably and validly account for this line pattern.

Recommendations:

I would suggest that:

- Fig 1 be downplayed significantly, as perhaps just a part of another figure.
- Statements up-playing the hypertension angle should be removed in the abstract, results and discussion.

Did you use live/dead staining in your flow cytometry? This is important because dead or dying cells can nonspecifically take up dyes. When you say non-specific binding sites were blocked, you should say how.

We thank the reviewer for highlighting this important point. In our experiments we did not use live/dead staining. However, to minimise differences in cell isolation between the groups, we always performed experiments using WT and TRPM7+/ Δ animals at the same time. For blocking the unspecific binding, we used normal rat serum 5% v/v in PBS. This information has been added in the current version of the article (revised manuscript, page 7, line 7; Supplementary text page 5; topic Flow cytometry, line 2-3)

I agree with Reviewer #1 in that the flow cytometry is not well done. That reviewer found the technique unconvincing. I find the data very simplistic, certainly not sufficient to make the claims regarding T-cells and macrophages as primary driving cardiac inflammation factors. There's no histology. There's no biochemical assays for inflammation. There's not a lot that I can see supporting this in the main manuscript. I don't think the inflammation work stands as is.

Recommendations:

I would suggest that:

- The flow should move to Supplementary.
- Fig. 9 should be modified to remove the macrophages, T-cells, and term inflammation. The strong data are just the molecular pathways, and should be limited to that.
- Mention of pro-inflammatory signaling be removed from the abstract, results, and significantly downplayed in the discussion.

Page 8: The sentence: "Plasma magnesium levels were significantly reduced in TRPM7+/ Δ kinase mice compared with WT controls (Table 2)", should read: "As baseline, plasma magnesium levels were significantly reduced in TRPM7+/ Δ kinase mice compared with WT controls (Table 2)."

We would like to thank the reviewer for his/her suggestion. We have made the appropriate changes in the main text (revised manuscript, page 9, paragraph 2, line 1-2)

OK

It is a shame that telemetry wasn't used to measure blood pressure in these animals.

We totally agree that telemetry is the gold standard for blood pressure measurement. However, this study was performed with many groups of animals, that would require numerous probes and platforms. This was not practical or feasible. In the present study we used plethysmography, a well established and optimized technique. To minimize errors and variation in blood pressure measurements, all animals were trained to the apparatus a week before experimentation. By day zero of the study, blood pressures were stabilized in each

animal. During the experimental period, blood pressure was measured twice weekly by the same person. To clarify this, we have provided additional information in the revised manuscript (revised manuscript, page 6, topic Blood pressure measurement).

I 100% agree with the Reviewer.

This should absolutely have been done. I don't think the blood pressure measurements are particularly valid when done this way in mice, and certainly not when the paper is framed around them. Labs studying hypertension use in vivo telemetry. They measure SBP, DBP, MABP. One more reason to remove the focus on blood pressure entirely from this paper, and definitely take Fig. 1 and relegate it to a much more minor observation.

Recommendations:

I would suggest that:

- As stated above, the focus on blood pressure be removed entirely from this paper
- Fig 1 be removed as a standalone figure and relegated to a more minor observation.
- Comments related to hypertension be removed from the abstract, and downplayed in the results and the discussion.

In the example Western in figure 2, the phosphorylation of TRPM7 in the M7+/ Δ mice is increased markedly by salt, but this is not reflected by the bar graph.

We agree with the reviewer that the images were not good representatives of the data. To address this issue, we extracted fresh renal protein lysates from all animals used in the different groups and performed new western blot studies. As a positive and negative control for the antibody, we used tissues from WT and TRPM7 kinase dead mice (TRPM7^{R/R}) that contain a K1646R point mutation and exhibit deficiency of the catalytic activity of TRPM7 kinase domain in the whole organism (Kaitsuka T, et al. Sci Rep. 2014 Jul 17;4:5718. Ryazanova LV et al. Scientific reports. 2014 Dec 23;4:7599) (revised manuscript, figure 2A, 2B, new data).

OK

On the first line of page 14, you say that TRPM7 activity is decreased by aldosterone-salt. Is this entirely based on its phosphorylation? In the referenced work it seems this is due to autophosphorylation. If the kinase dead enzyme really has an increase in phosphorylation in response to salt (see comment above), this wouldn't seem to be due to autophosphorylation. What upstream kinase might be involved.

We thank the reviewer for the observation and suggestions. TRPM7 contains a cytosolic α -kinase domain with an Mg²⁺/ATP-binding site, which is critical for the catalytic activity, and a point mutation on the lysine 1646 abolishes the kinase activity. Autophosphorylation of the TRPM7 kinase domain on serine 1511/1567 enhances kinase-substrate interactions leading to their serine/threonine phosphorylation (Matsushita M et al. J Biol Chem. 2005 27;280:20793-803). Animals that contain a K1646R point mutation exhibit deficiency of the catalytic activity of TRPM7 kinase domain in the whole organism and show reduced phosphorylation of the Ser1511 as demonstrated in this work (revised manuscript, figure 2A, new data) and by

others (Nat Commun. 2017 Dec 4;8(1):1917). We agree with the reviewer that the autophosphorylation may not be the only way to activate the kinase domain and that TRPM7 may also be a target for other serine/threonine kinases, that are still unknown. However, in our results we also found reduced TRPM7 total protein expression (revised manuscript, figure 2B; Discussion, page 15, first paragraph) in WT treated with aldosterone-salt and in TRPM7+/Δ mice. These new experiments were performed using an antibody specific to the residues 1146-1165, that are upstream to the truncated residue in TRPM7+/Δ mice (1538-1863). These new data show that WT animals treated with aldosterone-salt and TRPM7+/Δ mice exhibit both reduced TRPM7 channel expression and kinase phosphorylation.

OK

The most striking finding in table 2 is that the urinary sodium in response to salt and aldo is increased more than 4-fold (from 170 to 710) in the M7+/Δ mice. There are two issues here. First, without known sodium intake, this finding is impossible to interpret, but if these animals are at steady state, it indicates that they are consuming much more sodium than WT mice. The reason for this is unclear, but perhaps there is a central effect of the kinase portion of the enzyme.

We agree with the reviewer that the increased concentration of urinary sodium in TRPM7+/Δ mice treated with aldosterone-salt is a very interesting finding. We did not perform metabolic studies and accordingly can not comment on the exact salt intake for each animal. As the reviewer points out, our thesis is that the kinase domain of TRPM7+/Δ influences sodium metabolism, a possibility that we have addressed with new experiments.

To address the potential role of TRPM7 kinase on molecular processes that regulate sodium homeostasis, we focused on expression of major sodium transporters in the kidney, specifically ENaC and Na,K-ATPase. These are important regulators of renal sodium reabsorption and are modulated by aldosterone.

We found increased expression of ENaC induced by aldosterone-salt in both groups. However, Na,K-ATPase was increased only in WT mice treated with aldosterone-salt. Na,KATPase

is a recognised Mg²⁺ dependent enzyme expressed in the basolateral surface of the nephron. In physiologic concentrations of Mg²⁺, Na,K-ATPase pumps Na⁺ from the cytoplasm and takes up K⁺. Reduced intracellular [Na⁺] will force the Na⁺ gradient through the luminal surface, therefore increasing the Na⁺ reabsorption (Mayan H, et al., Physiol Rep. 2018;6:e13843).

These effects may be attenuated in conditions where Na,K-ATPase is downregulated as observed in TRPM7-deficient/hypomagnesemic mice. Mechanisms underlying these results are still unclear but might be dependent on reduced Mg²⁺ levels, since Na,K-ATPase (in fact all ATPases) are Mg²⁺-dependent enzymes. The new data have been added in Supplemental Figure 1A, 1B and in the discussion (revised manuscript, pages 15-16).

Findings from these studies suggest for the first time that renal sodium handling, is regulated by TRPM7-dependent mechanisms, possibly via altered sodium transport. This interesting paradigm requires further examination, with studies planned in the future as our mouse colonies grow.

Recommendations:

I would suggest that:

- I am OK with this explanation provided it is also actually written into the paper as a limitation of the study, with the references and conceptualization, into the manuscript itself, and not just as a rebuttal to the Reviewer offline.

*We would like to thank the reviewer and editor for the constructive comments. The paper has undergone further taking into account all suggestions.
We hope the paper is now acceptable for publication*

Our responses to each comment are highlighted in blue and italic.

Reviewer #1 (Remarks to the Author):

In this study, the authors describe the phenotype of M7+/ Δ mice, which have a mutation in the kinase domain of TRPM7, at baseline and in response to aldosterone, salt feeding or both. These animals seem to have augmented hypertension, impaired endothelium-dependent vascular relaxation and enhanced fibrosis, among other findings. While this is interesting, the paper largely comes across as a cataloguing of findings without establishing a fundamental cause of these abnormalities. Given that increased blood pressure might account for many of these findings, it would be interesting to normalize the blood pressure in the M7+/ Δ mice to see if this prevents the vascular and renal abnormalities.

The reviewer raises an important point. To dissect whether cardiovascular and renal changes are dependent or independent of augmented blood pressure responses in M7+/ Δ mice, we would have liked to perform new in vivo studies treating mice with agents that reduce blood pressure and/or normalise magnesium homeostasis. Unfortunately we have not been able to perform new animal experiments due to numerous challenges related to the Covid19 pandemic. Specifically, our labs were closed for many months and we needed to cull our mouse colonies. Since (partial) reopening of our labs, we have been attempting to re-derive our mouse colonies, but due to the M7+/ Δ mice being 'slow breeders', it is taking a long time to repopulate our colonies. Considering that we would need a large number of mice to perform the new studies as suggested by the reviewer (WT and M7+/ Δ mice treated with vehicle, aldosterone, salt, aldosterone+salt in the absence and presence of antihypertensive drugs and/or magnesium supplementation) we anticipate these studies will only be completed in 12-14 months time. Accordingly, while we would have liked to perform the new in vivo studies suggested by the reviewer, we are not able to do so at this time. Hopefully the reviewer will appreciate and understand these challenges. Nevertheless, we have attempted to answer the concerns raised using ex vivo and cell-based models.

Using an in vitro strategy, we performed new experiments to evaluate whether some of the changes observed in vivo can be ameliorated by magnesium. We showed that TRPM7-deficient animals exhibit increased sensitivity to blood pressure and cardiovascular fibrosis induced by aldosterone and salt. Because these animals exhibit hypomagnesemia, an important question to address is whether the pro-fibrotic effects are dependent on Mg²⁺, linked to TRPM7 deficiency. We sought to address this question using in vitro approaches.

Hearts from TRPM7+/ Δ mice treated with aldosterone and salt exhibit increased phosphorylation of ERK, Smad3, and Stat1 and expression of TGF β 1 and IL-6 (revised manuscript, Figure 7 and supplementary figure S5). Similar results were observed in primary culture of cardiac fibroblasts treated with aldosterone and salt. These effects were ameliorated by Mg²⁺ treatment, suggesting that at least part of these phenomena are Mg²⁺-dependent (revised manuscript, Figure 8 and supplementary figure S7).

At issue here is that the authors have framed the paper starting from the issue of hypertension. Given that they have done that, and it's in Fig. 1., it would seem reasonable of Reviewer #1 to request these additional studies. However, when I read the paper, I don't even think it's about hypertension, the framing is wrong. Since M7+aldo has high BP but not much fibrosis in the

heart (fig 5), it would seem to suggest to me that the high BP alone is not driving the pathology. I think the authors approach and explanations and new in vitro experiments are ok for this point. But the framing of the paper with Fig 1 starting with hypertension seems doesn't sit properly. If it's not that important, then it seems strange to start with it as Fig 1.

Recommendations:

I would suggest that:

- Fig 1 be downplayed as just a part of another figure.
- The statement in the abstract "aldosterone-mediated hypertension" be removed and read instead something like... "Our model revealed vascular dysfunction and cardiovascular-renal fibrosis were exaggerated in the TRPM7 het mice given aldosterone and salt." Similar changes should be made throughout the results and discussion.
- A supplementary picture should be added to show more frames like in Fig 5, especially the cardiac fibrosis, as one frame with bad fibrosis could be selective. How many hearts were actually processed for PicroSirius red? There appear to be 8 different samples in which collagen is measured for M7het aldo salt. Were there 8 hearts? Or did they just measure the same heart repeatedly. Given how important this image is to the study, it is imperative that they show multiple hearts and multiple areas of those hearts.

We thank the reviewer and editor for the excellent and insightful comments. Indeed, it is true that aldosterone has many organ and tissue effects that may be independent of hypertension. The multifunctional effects of aldosterone/salt are becoming more evident in cardiovascular, renal and metabolic pathophysiology. From a clinical viewpoint, aldosterone antagonists are being used for conditions other than hypertension.

Considering the excellent suggestion as highlighted above, we have followed the reviewer's suggestions and made appropriate changes in the paper. Blood pressure graphs are now combined with TRPM7 expression and phosphorylation in Figure 1C-1E. This format will highlight the importance of TRPM7 in the model. In addition, we changed the statement "aldosterone-mediated hypertension" and similar observations in the abstract, results and discussion to "cardiovascular and renal damage induced by aldosterone".

Regarding the fibrosis data, we randomly acquired 6-10 different images from each heart and kidneys. For aortas, we acquired 1-2 images. Collagen deposition was measured in every picture. Individual N value was obtained by the average measurement of all pictures from the same animal. As recommended, we added 4 additional pictures from cardiac tissues representing the data obtained from each animal in different groups in the supplemental information (Supp Figure 4A-4H). Additionally, we added representative figures from kidneys (Supp Figure 5A-5B) and aortas (Supp Figure 6A-6B) obtained from each animal used in the study

In figure 1, you should show the effects of aldo on WT and your kinase ko mice in the same plot and show us statistical analysis comparing these two. A better way to show this would be to show the effect of aldo only on WT and kinase deficient mice in one plot, effect of salt in another, and effect of salt + aldo in yet another. You have made these comparisons in supplemental table 2, but it's a shame to relegate these important comparisons to the supplemental section. Also in suppl table 2, isn't there a difference between kinase deficient and WT systolic BPs with aldo and salt? A symbol for this (‡) is lacking.

We thank the reviewer for the suggestion. We have reorganised the presentation of the graphs (revised manuscript figure 1A-C) and agree the data are easier to understand. The reviewer's observation regarding differences in blood pressure between aldo and aldo+salt M7+/ Δ mice is correct. We have added the symbol of significance, particularly at week 3 (‡)

(revised manuscript, figure 1C).

There are many issues with Fig. 1. in addition to the question of hypertension driving pathology, and the lack of telemetry data. Fig 1B is confusion (no difference except in 1 value at 1 timepoint – is this an anomaly? If not, what does it mean, if anything?). And Fig 1C is also confusing. What is happening at Week 3 here – why is one of the red lines going up temporarily and another one going down temporarily. What physiologic explanation would reasonably and validly account for this line pattern.

Recommendations:

I would suggest that:

- Fig 1 be downplayed significantly, as perhaps just a part of another figure.
- Statements up-playing the hypertension angle should be removed in the abstract, results and discussion.

We are in full agreement with the suggestions and accordingly have restructured the paper as suggested by the reviewer/editor (detailed above).

Summarizing the blood pressure data, it appears that TRPM7+/ Δ animals are more sensitive to blood pressure elevation, as observed in figure 1C. Salt treatment has effects on blood pressure only in TRPM7+/ Δ mice, week 4. This may relate to initial adaptive processes, which with time become maladaptive. It is possible that salt treatment for more than 4 weeks would show a consistently elevated blood pressure, as recently demonstrated by Kumagai et al (Hypertension. 2021 Jul;78(1):138-150.) who demonstrated that high salt and low Mg²⁺ diet increased blood pressure in 5 weeks. However, the dual effect of aldosterone and salt treatment likely has a more potent effect evidenced by blood pressure augmentation in TRPM7+/ Δ mice at week 3 compared to WT mice.

Taken together the data suggest that TRPM7+/ Δ mice are more sensitive to blood pressure elevation, which is treatment (aldo/salt)-dependent. These are interesting data, the mechanisms of which we will dissect in future studies.

As recommended, blood pressure data was changed to figure 1C-1E in the new version. Additionally, we removed statements “aldosterone-induced hypertension”, and focussed more on cardiovascular and renal fibrosis.

Did you use live/dead staining in your flow cytometry? This is important because dead or dying cells can nonspecifically take up dyes. When you say non-specific binding sites were blocked, you should say how.

We thank the reviewer for highlighting this important point. In our experiments we did not use live/dead staining. However, to minimise differences in cell isolation between the groups, we always performed experiments using WT and TRPM7+/ Δ animals at the same time. For blocking the unspecific binding, we used normal rat serum 5% v/v in PBS. This information has been added in the current version of the article (revised manuscript, page 7, line 7; Supplementary text page 5; topic Flow cytometry, line 2-3)

I agree with Reviewer #1 in that the flow cytometry is not well done. That reviewer found the technique unconvincing. I find the data very simplistic, certainly not sufficient to make the claims regarding T-cells and macrophages as primary driving cardiac inflammation factors. There's no histology. There's no biochemical assays for inflammation. There's not a lot that I can see supporting this in the main manuscript. I don't think the inflammation work stands as is.

Recommendations:

I would suggest that:

- The flow should move to Supplementary.
- Fig. 9 should be modified to remove the macrophages, T-cells, and term inflammation. The strong data are just the molecular pathways, and should be limited to that.
- Mention of pro-inflammatory signaling be removed from the abstract, results, and significantly downplayed in the discussion.

We thank the reviewer for these recommendations. All the flow cytometry data are now in the supplementary figures. Additionally, we modified Fig. 9 and changed the information about inflammatory response in the main text.

Page 8: The sentence: "Plasma magnesium levels were significantly reduced in TRPM7+/ Δ kinase mice compared with WT controls (Table 2)", should read: "As baseline, plasma magnesium levels were significantly reduced in TRPM7+/ Δ kinase mice compared with WT controls (Table 2)."

We would like to thank the reviewer for his/her suggestion. We have made the appropriate changes in the main text (revised manuscript, page 9, paragraph 2, line 1-2)

OK

It is a shame that telemetry wasn't used to measure blood pressure in these animals.

We totally agree that telemetry is the gold standard for blood pressure measurement. However, this study was performed with many groups of animals, that would require numerous probes and platforms. This was not practical or feasible. In the present study we used plethysmography, a well established and optimized technique. To minimize errors and variation in blood pressure measurements, all animals were trained to the apparatus a week before experimentation. By day zero of the study, blood pressures were stabilized in each animal. During the experimental period, blood pressure was measured twice weekly by the same person. To clarify this, we have provided additional information in the revised manuscript (revised manuscript, page 6, topic Blood pressure measurement).

I 100% agree with the Reviewer. This should absolutely have been done. I don't think the blood pressure measurements are particularly valid when done this way in mice, and certainly not when the paper is framed around them. Labs studying hypertension use in vivo telemetry. They measure SBP, DBP, MABP. One more reason to remove the focus on blood pressure entirely from this paper, and definitely take Fig. 1 and relegate it to a much more minor observation.

Recommendations:

I would suggest that:

- As stated above, the focus on blood pressure be removed entirely from this paper
- Fig 1 be removed as a standalone figure and relegated to a more minor observation.
- Comments related to hypertension be removed from the abstract, and downplayed in the results and the discussion.

As recommended, blood pressure data are now incorporated in Figs 1C-1D, together with TRPM7 expression and phosphorylation. Additionally, the text has been modified to be more focussed on cardiovascular and renal fibrosis.

In the example Western in figure 2, the phosphorylation of TRPM7 in the M7+/ Δ mice is increased markedly by salt, but this is not reflected by the bar graph.

We agree with the reviewer that the images were not good representatives of the data. To address this issue, we extracted fresh renal protein lysates from all animals used in the different groups and performed new western blot studies. As a positive and negative control for the antibody, we used tissues from WT and TRPM7 kinase dead mice (TRPM7^{R/R}) that contain a K1646R point mutation and exhibit deficiency of the catalytic activity of TRPM7 kinase domain in the whole organism (Kaitsuka T, et al. *Sci Rep.* 2014 Jul 17;4:5718. Ryazanova LV et al. *Scientific reports.* 2014 Dec 23;4:7599) (revised manuscript, figure 2A, 2B, new data).

OK

On the first line of page 14, you say that TRPM7 activity is decreased by aldo-salt. Is this entirely based on its phosphorylation? In the referenced work it seems this is due to autophosphorylation. If the kinase dead enzyme really has an increase in phosphorylation in response to salt (see comment above), this wouldn't seem to be due to autophosphorylation. What upstream kinase might be involved.

We thank the reviewer for the observation and suggestions. TRPM7 contains a cytosolic α -kinase domain with an Mg²⁺/ATP-binding site, which is critical for the catalytic activity, and a point mutation on the lysine 1646 abolishes the kinase activity. Autophosphorylation of the TRPM7 kinase domain on serine 1511/1567 enhances kinase-substrate interactions leading to their serine/threonine phosphorylation (Matsushita M et al. *J Biol Chem.* 2005 27;280:20793-803). Animals that contain a K1646R point mutation exhibit deficiency of the catalytic activity of TRPM7 kinase domain in the whole organism and show reduced phosphorylation of the Ser1511 as demonstrated in this work (revised manuscript, figure 2A, new data) and by others (*Nat Commun.* 2017 Dec 4;8(1):1917). We agree with the reviewer that the autophosphorylation may not be the only way to activate the kinase domain and that TRPM7 may also be a target for other serine/threonine kinases, that are still unknown. However, in our results we also found reduced TRPM7 total protein expression (revised manuscript, figure 2B; Discussion, page 15, first paragraph) in WT treated with aldosterone-salt and in TRPM7+/ Δ mice. These new experiments were performed using an antibody specific to the residues 1146-1165, that are upstream to the truncated residue in TRPM7+/ Δ mice (1538-1863). These new data show that WT animals treated with aldosterone-salt and TRPM7+/ Δ mice exhibit both reduced TRPM7 channel expression and kinase phosphorylation.

OK

The most striking finding in table 2 is that the urinary sodium in response to salt and aldo is increased more than 4-fold (from 170 to 710) in the M7+/ Δ mice. There are two issues here. First, without known sodium intake, this finding is impossible to interpret, but if these animals are at steady state, it indicates that they are consuming much more sodium than WT mice. The reason for this is unclear, but perhaps there is a central effect of the kinase portion of the enzyme.

We agree with the reviewer that the increased concentration of urinary sodium in TRPM7+/ Δ mice treated with aldosterone-salt is a very interesting finding. We did not perform metabolic studies and accordingly can not comment on the exact salt intake for each animal. As the reviewer points out, our thesis is that the kinase domain of TRPM7+/ Δ influences sodium metabolism, a possibility that we have addressed with new experiments. To address the potential role of TRPM7 kinase on molecular processes that regulate sodium homeostasis, we focused on expression of major sodium transporters in the kidney, specifically ENaC and Na,K-ATPase. These are important regulators of renal sodium reabsorption and are modulated by

aldosterone. We found increased expression of ENaC induced by aldosterone-salt in both groups. However, Na,K-ATPase was increased only in WT mice treated with aldosterone-salt. Na,K-ATPase is a recognised Mg²⁺ dependent enzyme expressed in the basolateral surface of the nephron. In physiologic concentrations of Mg²⁺, Na,K-ATPase pumps Na⁺ from the cytoplasm and takes up K⁺. Reduced intracellular [Na⁺] will force the Na⁺ gradient through the luminal surface, therefore increasing the Na⁺ reabsorption (Mayan H, et al., *Physiol Rep.* 2018;6:e13843). These effects may be attenuated in conditions where Na,K-ATPase is downregulated as observed in TRPM7-deficient/hypomagnesemic mice. Mechanisms underlying these results are still unclear but might be dependent on reduced Mg²⁺ levels, since Na,K-ATPase (in fact all ATPases) are Mg²⁺-dependent enzymes. The new data have been added in Supplemental Figure 1A, 1B and in the discussion (revised manuscript, pages 15-16). Findings from these studies suggest for the first time that renal sodium handling, is regulated by TRPM7-dependent mechanisms, possibly via altered sodium transport. This interesting paradigm requires further examination, with studies planned in the future as our mouse colonies grow.

Recommendations:

I would suggest that:

- I am OK with this explanation provided it is also actually written into the paper as a limitation of the study, with the references and conceptualization, into the manuscript itself, and not just as a rebuttal to the Reviewer offline.

We thank the reviewer for recommendations. We added the new information in the discussion. Firstly, we mentioned the study limitation (line 377-379) and thereafter in the same paragraph we added the information about Na⁺ homeostasis (line 385-388).